# Fast Rates of ERM and Stochastic Approximation: Adaptive to Error Bound Conditions

**Mingrui Liu[†], Xiaoxuan Zhang[†], Lijun Zhang[‡], Rong Jin[♮], Tianbao Yang[†]**
[†]Department of Computer Science, The University of Iowa, Iowa City, IA 52242, USA
[‡]National Key Laboratory for Novel Software Technology, Nanjing University, China
[♮] Machine Intelligence Technology, Alibaba Group, Bellevue, WA 98004, USA
`mingrui-liu@uiowa.edu, zljzju@gmail.com, tianbao-yang@uiowa.edu`

## Abstract

Error bound conditions (EBC) are properties that characterize the growth of an objective function when a point is moved away from the optimal set. They have recently received increasing attention for developing optimization algorithms with fast convergence. However, the studies of EBC in statistical learning are hitherto still limited. The main contributions of this paper are two-fold. First, we develop fast and intermediate rates of empirical risk minimization (ERM) under EBC for risk minimization with Lipschitz continuous, and smooth convex random functions. Second, we establish fast and intermediate rates of an efficient stochastic approximation (SA) algorithm for risk minimization with Lipschitz continuous random functions, which requires only one pass of $n$ samples and adapts to EBC. For both approaches, the convergence rates span a full spectrum between $\widetilde{O}(1/\sqrt{n})$ and $\widetilde{O}(1/n)$ depending on the power constant in EBC, and could be even faster than $O(1/n)$ in special cases for ERM. Moreover, these convergence rates are automatically adaptive without using any knowledge of EBC.

## 1 Introduction

In this paper, we focus on the following stochastic convex optimization problems arising in statistical learning and many other fields:

$$\min_{\mathbf{w} \in \mathcal{W}} P(\mathbf{w}) \triangleq \mathbb{E}_{\mathbf{z} \sim \mathbb{P}}[f(\mathbf{w}, \mathbf{z})], \tag{1}$$

and more generally

$$\min_{\mathbf{w} \in \mathcal{W}} P(\mathbf{w}) \triangleq \mathbb{E}_{\mathbf{z} \sim \mathbb{P}}[f(\mathbf{w}, \mathbf{z})] + r(\mathbf{w}), \tag{2}$$

where $f(\cdot, \mathbf{z}) : \mathcal{W} \to \mathbb{R}$ is a random function depending on a random variable $\mathbf{z} \in \mathcal{Z}$ that follows a distribution $\mathbb{P}$, $r(\mathbf{w})$ is a lower semi-continuous convex function. In statistical learning [48], the problem above is also referred to as **risk minimization** where $\mathbf{z}$ is interpreted as data, $\mathbf{w}$ is interpreted as a model (or hypothesis), $f(\cdot, \cdot)$ is interpreted as a loss function, and $r(\cdot)$ is a regularization. For example, in supervised learning one can take $\mathbf{z} = (\mathbf{x}, y)$ - a pair of feature vector $\mathbf{x} \in \mathcal{X} \subseteq \mathbb{R}^d$ and label $\mathbf{y} \in \mathcal{Y}$, $f(\mathbf{w}, \mathbf{z}) = \ell(\mathbf{w}(\mathbf{x}), y)$ - a loss function measuring the error of the prediction $\mathbf{w}(\mathbf{x}) : \mathcal{X} \to \mathcal{Y}$ made by the model $\mathbf{w}$. Nonetheless, we emphasize that the risk minimization problem (1) is more general than supervised learning and could be more challenging (c.f. [35]). In this paper, we assume that $\mathcal{W} \subseteq \mathbb{R}^d$ is a compact and convex set. Let $\mathcal{W}_* = \arg\min_{\mathbf{w} \in \mathcal{W}} P(\mathbf{w})$ denote the optimal set and $P_* = \min_{\mathbf{w} \in \mathcal{W}} P(\mathbf{w})$ denote the optimal risk.

There are two popular approaches for solving the risk minimization problem. The first one is by empirical risk minimization that minimizes the empirical risk defined over a set of $n$ i.i.d. samples drawn from the same distribution $\mathbb{P}$ (sometimes with a regularization term on the model). The second

approach is called stochastic approximation that iteratively learns the model from random samples $\mathbf{z}_t \sim \mathbb{P}, t = 1, \ldots, n$. Both approaches have been studied broadly and extensive results are available about the theoretical guarantee of the two approaches in the machine learning and optimization community. A central theme in these studies is to bound the excess risk (or optimization error) of a learned model $\widehat{\mathbf{w}}$ measured by $P(\widehat{\mathbf{w}}) - P_*$, i.e., given a set of $n$ samples $(\mathbf{z}_1, \ldots, \mathbf{z}_n)$ how fast the learned model converges to the optimal model in terms of the excess risk.

A classical result about the excess risk bound for the considered risk minimization problem is in the order of $\widetilde{O}(\sqrt{d/n})$ [1] and $O(\sqrt{1/n})$ for ERM and SA, respectively, under appropriate conditions of the loss functions (e.g., Lipschitz continuity, convexity) [29, 35]. Various studies have attempted to establish faster rates by imposing additional conditions on the loss functions (e.g., strong convexity, smoothness, exponential concavity) [13, 42, 21], or on both the loss functions and the distribution (e.g., Tsybakov condition, Bernstein condition, central condition) [45, 3, 46]. In this paper, we will study a different family of conditions called the error bound conditions (EBC) (see Definition 1), which has a long history in the community of optimization and variational analysis [31] and recently revived for developing fast optimization algorithms without strong convexity [4, 6, 17, 28, 54]. However, the exploration of EBC in statistical learning for risk minimization is still under-explored and the connection to other conditions is not fully understood.

**Definition 1.** *For any* $\mathbf{w} \in \mathcal{W}$, *let* $\mathbf{w}^* = \arg\min_{\mathbf{u} \in \mathcal{W}_*} \|\mathbf{u} - \mathbf{w}\|_2$ *denote an optimal solution closest to* $\mathbf{w}$, *where* $\mathcal{W}_*$ *is the set containing all optimal solutions. Let* $\theta \in (0, 1]$ *and* $0 < \alpha < \infty$. *The problem (1) satisfies an EBC$(\theta, \alpha)$ if for any* $\mathbf{w} \in \mathcal{W}$, *the following inequality holds*

$$\|\mathbf{w} - \mathbf{w}^*\|_2^2 \leq \alpha (P(\mathbf{w}) - P(\mathbf{w}^*))^\theta. \tag{3}$$

This condition has been well studied in optimization and variational analysis. Many results are available for understanding the condition for different problems. For example, it has been shown that when $P(\mathbf{w})$ is semi-algebraic and continuous, the inequality (3) is known to hold on any compact set with certain $\theta \in (0, 1]$ and $\alpha > 0$ [4] [2]. We will study both ERM and SA under the above error bound condition. In particular, we show that the benefits of exploiting EBC in statistical learning are noticeable and profound by establishing the following results.

- **Result I.** First, we show that for Lipchitz continous loss EBC implies a *relaxed* Bernstein condition, and therefore leads to intermediate rates of $\widetilde{O}((d/n)^{\frac{1}{2-\theta}})$ for Lipschitz continuous loss. Although this result does not improve over existing rates based on Bernstein condition, however, we emphasize that it provides an alternative route for establishing fast rates and brings richer results than literature to statistical learning in light of the examples provided in this paper.
- **Result II.** Second, we develop fast and optimistic rates of ERM for non-negative, Lipschitz continuous and smooth convex loss functions in the order of $\widetilde{O}(d/n + (dP_*/n)^{\frac{1}{2-\theta}})$, and in the order of $\widetilde{O}((d/n)^{\frac{2}{2-\theta}} + (dP_*/n)^{\frac{1}{2-\theta}})$ when the sample size $n$ is sufficiently large, which imply that when the optimal risk $P_*$ is small one can achieve a fast rate of $\widetilde{O}(d/n)$ even with $\theta < 1$ and a faster rate of $\widetilde{O}((d/n)^{\frac{2}{2-\theta}})$ when $n$ is sufficiently large.
- **Result III.** Third, we develop an efficient SA algorithm with almost the same per-iteration cost as stochastic subgradient methods for Lipschitz continuous loss, which achieves the same order of rate $\widetilde{O}((1/n)^{\frac{1}{2-\theta}})$ as ERM without an explicit dependence on $d$. More importantly it is "parameter"-free with no need of prior knowledge of $\theta$ and $\alpha$ in EBC.

Overall, these results not only strengthen the understanding of ERM for statistical learning but also bring new fast stochastic algorithms for solving a broad range of statistical learning problems. Before ending this section, we would like to point out that all the results are automatically adaptive to the largest possible value of $\theta \in (0, 1]$ in hindsight of the problem, and the dependence on $d$ for ERM is generally unavoidable according to the lower bounds studied in [9].

## 2  Related Work

The results for statistical learning under EBC are limited. A similar one to our **Result I** for ERM was established in [39]. However, their result requires the convexity condition of random loss functions,

making it weaker than our result. Ramdas and Singh [33] and Xu et al. [50] considered SA under the EBC condition and established similar adaptive rates. Nonetheless, their stochastic algorithms require knowing the values of $\theta$ and possibly the constant $\alpha$ in the EBC. In contrast, the SA algorithm in this paper is "parameter"-free without the need of knowing $\theta$ and $\alpha$ while still achieving the adaptive rates of $O(1/n^{2-\theta})$. Fast rates under strong convexity (a special case of EBC) are well-known for ERM, online optimization and SA [35, 43, 13, 16, 36, 14]. In the presence of strong convexity of $P(\mathbf{w})$, our results of ERM and SA recover known rates (see below for more discussions).

Fast (intermediate) rates of ERM have been studied under various conditions, including Tsybakov margin condition [44, 25], Bernstein condition [3, 2, 19], exp-concavity condition [21, 11, 26, 51], mixability condition [27], central condition [46], etc. The Bernstein condition (see Definition 2) is a generalization of Tsybakov margin condition for classification. The connection between the exp-concavity condition, the Bernstein condition and the $v$-central condition was studied in [46]. In particular, the exp-concavity implies a $v$-central condition under an appropriate condition of the decision set $\mathcal{W}$ (e.g., well-specificity or convexity). With the bounded loss condition, the Bernstein condition implies the $v$-central condition and the $v$-central condition also implies a Bernstein condition.

In this work, we also study the connection between the EBC and the Bernstein condition and the $v$-central condition. In particular, we will develop weaker forms of the Bernstein condition and the $v$-central condition from the EBC for Lipschitz continuous loss functions. Building on this connection, we establish our **Result I**, which is on a par with existing results for bounded loss functions relying on the Bernstein condition or the central condition. Nevertheless, we emphasize that employing the EBC for developing fast rates has noticeable benefits: (i) it is complementary to the Bernstein condition and the central condition and enjoyed by several interesting problems whose fast rates are not exhibited yet; (ii) it can be leveraged for developing fast and intermediate optimistic rates for non-negative and smooth loss functions; (iii) it can be leveraged to develop efficient SA algorithms with intermediate and fast convergence rates.

Sebro et al. [42] established an optimistic rate of $O(1/n + \sqrt{P_*/n})$ of both ERM and SA for supervised learning with generalized linear loss functions. However, their SA algorithm requires knowing the value of $P_*$. Recently, Zhang et al. [55] considered the general stochastic optimization problem (1) with non-negative and smooth loss functions and achieved a series of optimistic results. It is worth mentioning that their excess risk bounds for both convex problems and strongly convex problems are special cases of our **Result II** when $\theta = 0$ and $\theta = 1$, respectively. However, the intermediate optimistic rates for $\theta \in (0, 1)$ are first shown in this paper. Importantly, our **Result II** under the EBC with $\theta = 1$ is more general than the result in [55] under strong convexity assumption.

Finally, we discuss about stochastic approximation algorithms with fast and intermediate rates to understand the significance of our **Result III**. Different variants of stochastic gradient methods have been analyzed for stochastic strongly convex optimization [14, 32, 38] with a fast rate of $O(1/n)$. But these stochastic algorithms require knowing the strong convexity modulus. A recent work established adaptive regret bounds $O(n^{\frac{1-\theta}{2-\theta}})$ for online learning with a total of $n$ rounds under the Bernstein condition [20]. However, their methods are based on the second-order methods and therefore are not as efficient as our stochastic approximation algorithm. For example, for online convex optimization they employed the MetaGrad algorithm [47], which needs to maintain $\log(n)$ copies of the online Newton step (ONS) [13] with different learning rates. Notice that the per-iteration cost of ONS is usually $O(d^4)$ even for very simple domain $\mathcal{W}$ [21], while that of our SA algorithm is dominated by the Euclidean projection onto $\mathcal{W}$ that is as fast as $O(d)$ for a simple domain.

## 3 Empirical Risk Minimization (ERM)

We first formally state the minimal assumptions that are made throughout the paper. Additional assumptions will be made in the sequel for developing fast rates for different families of the random functions $f(\mathbf{w}, \mathbf{z})$.

**Assumption 1.** *For the stochastic optimization problems (1) and (2), we assume: (i) $P(\mathbf{w})$ is a convex function, $\mathcal{W}$ is a closed and bounded convex set, i.e., there exists $R > 0$ such that $\|\mathbf{w}\|_2 \leq R$ for any $\mathbf{w} \in \mathcal{W}$, and $r(\mathbf{w})$ is a Lipschitz continuous convex function. (ii) the problem (1) and (2) satisfy an EBC($\theta, \alpha$), i.e., there exist $\theta \in (0, 1]$ and $0 < \alpha < \infty$ such that the inequality (3) hold.*

In this section, we focus on the development of theory of ERM for risk minimization. In particular, we learn a model $\widehat{\mathbf{w}}$ by solving the following ERM problem corresponding to (1):

$$\widehat{\mathbf{w}} \in \arg \min_{\mathbf{w} \in \mathcal{W}} P_n(\mathbf{w}) \triangleq \frac{1}{n} \sum_{i=1}^{n} f(\mathbf{w}, \mathbf{z}_i) \tag{4}$$

where $\mathbf{z}_1, \ldots, \mathbf{z}_n$ are i.i.d samples following the distribution $\mathbb{P}$. A similar ERM problem can be formulated for (2). This section is divided into two subsections. First, we establish intermediate rates of ERM under EBC when the random function is Lipschitz continuous. Second, we develop intermediate rates of ERM under EBC when the random function is smooth. In the sequel and the supplement, we use $\vee$ to denote the max operation and use $\wedge$ to denote the min operation.

## 3.1 ERM for Lipschitz continuous random functions

In this subsection, w.l.o.g we restrict our attention to (1) since we make the following assumption besides Assumption 1. If $r(\mathbf{w})$ is present, it can be absorbed into $f(\mathbf{w}, \mathbf{z})$.

**Assumption 2.** *For the stochastic optimization problem (1), we assume that $f(\mathbf{w}, \mathbf{z})$ is a $G$-Lipschitz continuous function w.r.t $\mathbf{w}$ for any $\mathbf{z} \in \mathcal{Z}$.*

It is notable that we do not assume $f(\mathbf{w}, \mathbf{z})$ is convex in terms of $\mathbf{w}$ or any $\mathbf{z}$. First, we compare EBC with two very important conditions considered in literature for developing fast rates of ERM, namely the Bernstein condition and the central condition. We first give the definitions of these two conditions.

**Definition 2.** *(Bernstein Condition) Let $\beta \in (0, 1]$ and $B \geq 1$. Then $(f, \mathbb{P}, \mathcal{W})$ satisfies the $(\beta, B)$-Bernstein condition if there exists a $\mathbf{w}_* \in \mathcal{W}$ such that for any $\mathbf{w} \in \mathcal{W}$*

$$\mathbb{E}_{\mathbf{z}}[(f(\mathbf{w}, \mathbf{z}) - f(\mathbf{w}_*, \mathbf{z}))^2] \leq B(\mathbb{E}_{\mathbf{z}}[f(\mathbf{w}, \mathbf{z}) - f(\mathbf{w}_*, \mathbf{z})])^{\beta}. \tag{5}$$

It is clear that if such an $\mathbf{w}_*$ exists it has to be the minimizer of the risk.

**Definition 3.** *($v$-Central Condition) Let $v : [0, \infty) \to [0, \infty)$ be a bounded, non-decreasing function satisfying $v(x) > 0$ for all $x > 0$. We say that $(f, \mathbb{P}, \mathcal{W})$ satisfies the $v$-central condition if for all $\varepsilon \geq 0$, there exists $\mathbf{w}_* \in \mathcal{W}$ such that for any $\mathbf{w} \in \mathcal{W}$ the following holds with $\eta = v(\varepsilon)$.*

$$\mathbb{E}_{\mathbf{z} \sim \mathbb{P}} \left[ e^{\eta(f(\mathbf{w}_*, \mathbf{z}) - f(\mathbf{w}, \mathbf{z}))} \right] \leq e^{\eta \varepsilon}. \tag{6}$$

If $v(\varepsilon)$ is a constant for all $\varepsilon \geq 0$, the $v$-central condition reduces to the strong $\eta$-central condition, which implies the $O(1/n)$ fast rate [46]. The connection between the Bernstein condition or $v$-central condition has been studied in [46]. For example, if the random functions $f(\mathbf{w}, \mathbf{z})$ take values in $[0, a]$, then $(\beta, B)$-Bernstein condition implies $v$-central condition with $v(x) \propto x^{1-\beta}$.

The following lemma shows that for Lipchitz continuous function, EBC condition implies a relaxed Bernstein condition and a relaxed $v$-central condition.

**Lemma 1.** *(**Relaxed Bernstein condition and $v$-central condition**) Suppose Assumptions 1, 2 hold. For any $\mathbf{w} \in \mathcal{W}$, there exists $\mathbf{w}^* \in \mathcal{W}_*$ (which is actually the one closest to $\mathbf{w}$), such that*

$$\mathbb{E}_{\mathbf{z}}[(f(\mathbf{w}, \mathbf{z}) - f(\mathbf{w}^*, \mathbf{z}))^2] \leq B(\mathbb{E}_{\mathbf{z}}[f(\mathbf{w}, \mathbf{z}) - f(\mathbf{w}^*, \mathbf{z})])^{\theta},$$

*where $B = G^2 \alpha$, and $\mathbb{E}_{\mathbf{z} \sim \mathbb{P}} \left[ e^{\eta(f(\mathbf{w}^*, \mathbf{z}) - f(\mathbf{w}, \mathbf{z}))} \right] \leq e^{\eta \varepsilon}$, where $\eta = v(\varepsilon) := c\varepsilon^{1-\theta} \wedge b$. Additionally, for any $\varepsilon > 0$ if $P(\mathbf{w}) - P(\mathbf{w}^*) \geq \varepsilon$, we have $\mathbb{E}_{\mathbf{z} \sim \mathbb{P}} \left[ e^{v(\varepsilon)(f(\mathbf{w}^*, \mathbf{z}) - f(\mathbf{w}, \mathbf{z}))} \right] \leq 1$, where $b > 0$ is any constant and $c = 1/(\alpha G^2 \kappa(4GRb))$, where $\kappa(x) = (e^x - x - 1)/x^2$.*

**Remark:** There is a subtle difference between the above relaxed Bernstein condition and $v$-central condition and their original definitions in Definitions 2 and 3. The difference is that in Definitions 2 and 3, it requires there exists a universal $\mathbf{w}_*$ for all $\mathbf{w} \in \mathcal{W}$ such that (5) and (6) hold. In Lemma 1 it only requires for every $\mathbf{w} \in \mathcal{W}$ there exists one $\mathbf{w}^*$ that could be different for different $\mathbf{w}$ such that (5) and (6) hold. This relaxation enables us to establish richer results by exploring EBC than the Bernstein condition and $v$-central condition, which are postponed to Section 5.

Next, we present the main result of this subsection.

**Theorem 1 (Result I).** *Suppose Assumptions 1, 2 hold. For any $n \geq aC$, with probability at least $1 - \delta$ we have*

$$P(\widehat{\mathbf{w}}) - P_* \leq O\left(\frac{d \log n + \log(1/\delta)}{n}\right)^{\frac{1}{2-\theta}}, \tag{7}$$

*where $a = 3(d \log(32GRn^{1/(2-\theta)}) + \log(1/\delta))/c + 1$ and $C > 0$ is some constant.*

**Remark:** The proof utilizes Lemma 1 and follows similarly as the proofs in previous studies [46, 26] based on $v$-central condition. Our analysis essentially shows that relaxed Bernstein condition and relaxed $v$-central condition with non-universal $\mathbf{w}^*$ suffice to establish the intermediate rates. Although the rate in Theorem 1 does not improve that in previous works [46], the relaxation brought by EBC allows us to establish fast rates for interesting problems that were unknown before. More details are postponed into Section 5. For example, under the condition that the input data $\mathbf{x}, y$ are bounded, ERM for hinge loss minimization with $\ell_1, \ell_\infty$ norm constraints, and for minimizing a quadratic function and an $\ell_1$ norm regularization enjoys an $\widetilde{O}(1/n)$ fast rate. To the best of our knowledge, such a fast rate of ERM for these problems has not been shown in literature using other conditions or theories.

## 3.2 ERM for non-negative, Lipschitz continuous and smooth convex random functions

Below we will present improved optimistic rates of ERM for non-negative smooth loss functions expanding the results in [55]. To be general, we consider (2) and the following ERM problem:

$$\widehat{\mathbf{w}} \in \arg\min_{\mathbf{w} \in \mathcal{W}} P_n(\mathbf{w}) \triangleq \frac{1}{n} \sum_{i=1}^{n} f(\mathbf{w}, \mathbf{z}_i) + r(\mathbf{w}) \tag{8}$$

Besides Assumptions 1, 2, we further make the following assumption for developing faster rates.

**Assumption 3.** *For the stochastic optimization problem (1), we assume $f(\mathbf{w}, \mathbf{z})$ is a non-negative and $L$-smooth convex function w.r.t $\mathbf{w}$ for any $\mathbf{z} \in \mathcal{Z}$.*

It is notable that we do not assume that $r(\mathbf{w})$ is smooth. Our main result in this subsection is presented in the following theorem.

**Theorem 2 (Result II).** *Under Assumptions 1, 2, and 3, with probability at least $1 - \delta$ we have*

$$P(\widehat{\mathbf{w}}) - P_* \leq O\left(\frac{d \log n + \log(1/\delta)}{n} + \left[\frac{(d \log n + \log(1/\delta))P_*}{n}\right]^{\frac{1}{2-\theta}}\right).$$

*When $n \geq \Omega\left(\left(\alpha^{1/\theta} d \log n\right)^{2-\theta}\right)$, with probability at least $1 - \delta$,*

$$P(\widehat{\mathbf{w}}) - P_* \leq O\left(\left[\frac{d \log n + \log(1/\delta)}{n}\right]^{\frac{2}{2-\theta}} + \left[\frac{(d \log n + \log(1/\delta))P_*}{n}\right]^{\frac{1}{2-\theta}}\right).$$

**Remark:** The constant in big $O$ and $\Omega$ can be seen from the proof, which is tedious and included in the supplement. Here we focus on the understanding of the results. First, the above results are optimistic rates that are no worse than that in Theorem 1. Second, the first result implies that when the optimal risk $P_*$ is less than $O((d \log n/n)^{1-\theta})$, the excess risk bound is in the order of $O(d \log n/n)$. Third, when the number of samples $n$ is sufficiently large and the optimal risk is sufficiently small, the second result can imply a faster rate than $O(d \log n/n)$. Considering smooth functions presented in Section 5 with $\theta = 1$, when $n \geq \Omega(\alpha d \log n)$ and $P_* \leq O(d \log n/n)$ (large-sample and small optimal risk), the excess risk can be bounded by $O((d \log n/n)^2)$. In another word, the sample complexity for achieving an $\epsilon$-excess risk bound is given by $\widetilde{O}(d/\sqrt{\epsilon})$. To the best of our knowledge, the sample complexity of ERM in the order of $1/\sqrt{\epsilon}$ for these examples is the first result appearing in the literature.

## 4  Efficient SA for Lipschitz continuous random functions

In this section, we will present intermediate rates of an efficient stochastic approximation algorithm for solving (1) adaptive to the EBC under the Assumption 1 and 2. Note that (2) can be considered as a special case by absorbing $r(\mathbf{w})$ into $f(\mathbf{w}, \mathbf{z})$.

| **Algorithm 1** SSG($\mathbf{w}_1, \gamma, T, \mathcal{W}$) | **Algorithm 2** ASA($\mathbf{w}_1, n, R$) |
|---|---|
| **Input:** $\mathbf{w}_1 \in \mathcal{W}, \gamma > 0$ and $T$ | 1: Set $R_0 = 2R, \widehat{\mathbf{w}}_0 = \mathbf{w}_1, m = \lfloor \frac{1}{2} \log_2 \frac{2n}{\log_2 n} \rfloor - 1, n_0 = \lfloor \frac{n}{m} \rfloor$ |
| 1: **for** $t = 1, \dots, T$ **do** | 2: **for** $k = 1, \dots, m$ **do** |
| 2: $\quad \mathbf{w}_{t+1} = \Pi_{\mathcal{W}}(\mathbf{w}_t - \gamma g_t)$ | 3: $\quad$ Set $\gamma_k = \frac{R_{k-1}}{G\sqrt{n_0+1}}$ and $R_k = R_{k-1}/2$ |
| 3: **end for** | 4: $\quad \widehat{\mathbf{w}}_k = $ SSG($\widehat{\mathbf{w}}_{k-1}, \gamma_k, n_0, \mathcal{W} \cap \mathcal{B}(\widehat{\mathbf{w}}_{k-1}, R_{k-1})$) |
| 4: $\widehat{\mathbf{w}}_T = \frac{1}{T+1} \sum_{t=1}^{T+1} \mathbf{w}_t$ | 5: **end for** |
| 5: return $\widehat{\mathbf{w}}_T$ | 6: return $\widehat{\mathbf{w}}_m$ |

Denote by $\mathbf{z}_1, \dots \mathbf{z}_k, \dots$ i.i.d samples drawn sequentially from the distribution $\mathbb{P}$, by $g_k \in \partial f(\mathbf{w}, \mathbf{z}_k)|_{\mathbf{w}=\mathbf{w}_k}$ a *stochastic subgradient* evaluated at $\mathbf{w}_k$ with sample $\mathbf{z}_k$, and by $\mathcal{B}(\mathbf{w}, R)$ a bounded ball centered at $\mathbf{w}$ with a radius $R$. By the Lipschitz continuity of $f$, we have $\|\partial f(\mathbf{w}, \mathbf{z})\|_2 \leq G$ for $\forall \mathbf{w} \in \mathcal{W}, \forall \mathbf{z} \in \mathcal{Z}$.

The proposed adaptive stochastic approximation algorithm is presented in Algorithm 2, which is referred to as ASA. The updates are divided into $m$ stages, where at each stage a stochastic subgradient method (Algorithm 1) is employed for running $n_0 = \lfloor n/m \rfloor$ iterations with a constant step size $\gamma_k$. The step size $\gamma_k$ will be decreased by half after each stage and the next stage will be warm-started using the solution returned from the last stage as the initial solution. The projection onto the intersection of $\mathcal{W}$ and a shrinking bounded ball at each stage is a commonly used trick for the high probability analysis [14, 15, 49]. We emphasize that the subroutine in ASA can be replaced by other SA algorithms, e.g., the proximal variant of stochastic subgradient for handling a non-smooth deterministic component such as $\ell_1$ norm regularization [7], stochastic mirror descent with with a $p$-norm divergence function [8], and etc. Please see an example in the supplement.

It is worth mentioning that the dividing schema of ASA is due to [15], which however restricts its analysis to uniformly convex functions where uniform convexity is a stronger condition than the EBC. ASA is also similar to a recently proposed accelerated stochastic subgradient (ASSG) method under the EBC [49]. However, the key differences are that (i) ASA is developed for a fixed number of iterations while ASSG is developed for a fixed accuracy level $\epsilon$; (ii) the adaptive iteration complexity of ASSG requires knowing the value of $\theta \in (0, 2]$ while ASA does not require the value of $\theta$. As a trade-off, we restrict our attention to $\theta \in (0, 1]$.

**Theorem 3** (**Result III**). *Suppose Assumptions 1 and 2 hold, and $\|\mathbf{w}_1 - \mathbf{w}^*\|_2 \leq R_0$, where $\mathbf{w}^*$ is the closest optimal solution to $\mathbf{w}_1$. Define $\bar{\alpha} = \max(\alpha G^2, (R_0 G)^{2-\theta})$. For $n \geq 100$ and any $\delta \in (0, 1)$, with probability at least $1 - \delta$, we have*

$$P(\widehat{\mathbf{w}}_m) - P_* \leq O\left(\frac{\bar{\alpha}(\log(n)\log(\log(n)/\delta))}{n}\right)^{\frac{1}{2-\theta}}.$$

**Remark:** The significance of the result is that although Algorithm 2 does not utilize any knowledge about EBC, it is automatically adaptive to the EBC. As a final note, the projection onto the intersection of $\mathcal{W}$ and a bounded ball can be efficiently computed by employing the projection onto $\mathcal{W}$ and a binary search for the Lagrangian multiplier of the ball constraint. Moreover, we can replace the subroutine with a slightly different variant of SSG to get around of the projection onto the intersection of $\mathcal{W}$ and a bounded ball, which is presented in the supplement.

## 5 Applications

From the last two sections, we can see that $\theta = 1$ is a favorable case, which yields the fastest rate in our results. It is obvious that if $f(\mathbf{w}, \mathbf{z})$ is strongly convex or $P(\mathbf{w})$ is strongly convex, then EBC($\theta = 1, \alpha$) holds. Below we show some examples of problem (1) and (2) with $\theta = 1$ without strong convexity, which not only recover some known results of fast rate $\widetilde{O}(d/n)$, but also induce new results of fast rates that are even faster than $\widetilde{O}(d/n)$.

**Quadratic Problems (QP):** $\quad \min_{\mathbf{w} \in \mathcal{W}} P(\mathbf{w}) \triangleq \mathbf{w}^\top \mathbb{E}_{\mathbf{z}}[A(\mathbf{z})]\mathbf{w} + \mathbf{w}^\top \mathbb{E}_{\mathbf{z}'}[\mathbf{b}(\mathbf{z}')] + c \qquad (9)$

where $c$ is a constant. The random function can be taken as $f(\mathbf{w}, \mathbf{z}, \mathbf{z}') = \mathbf{w}^\top A(\mathbf{z})\mathbf{w} + \mathbf{w}^\top \mathbf{b}(\mathbf{z}') + c$. We have the following corollary.

**Corollary 1.** *If $\mathbb{E}_{\mathbf{z}}[A(\mathbf{z})]$ is a positive semi-definite matrix (not necessarily positive definite) and $\mathcal{W}$ is a bounded polyhedron, then the problem (9) satisfies $EBC(\theta = 1, \alpha)$. Assume that $\max(\|A(\mathbf{z})\|_2, \|b(\mathbf{z}')\|_2) \leq \sigma < \infty$, then ERM has a fast rate at least $\widetilde{O}(d/n)$. If $f(\mathbf{w}, \mathbf{z}, \mathbf{z}')$ is further non-negative, convex and smooth, then ERM has a fast rate of $\widetilde{O}((d/n)^2 + dP_*/n)$ when $n \geq \Omega(d \log n)$. ASA has a convergence rate of $\widetilde{O}(1/n)$.*

Next, we present some instances of the quadratic problem (9).
*Instance 1 of QP: minimizing the expected square loss.* Consider the following problem:

$$\min_{\mathbf{w} \in \mathcal{W}} P(\mathbf{w}) \triangleq \mathbb{E}_{\mathbf{x}, y}[(\mathbf{w}^\top \mathbf{x} - y)^2] \tag{10}$$

where $\mathbf{x} \in \mathcal{X}, y \in \mathcal{Y}$ and $\mathcal{W}$ is a bounded polyhedron (e.g., $\ell_1$-ball or $\ell_\infty$-ball). It is not difficult to show that it is an instance of (9) and has the property that $f(\mathbf{w}, \mathbf{z}, \mathbf{z}')$ is non-negative, smooth, convex, Lipchitz continuous over $\mathcal{W}$. The convergence results in Corollary 1 for this instance not only recover some known results of $\widetilde{O}(d/n)$ rate [22, 26], but also imply a faster rate than $\widetilde{O}(d/n)$ in a large-sample regime and an optimistic case when $n \geq \Omega(d \log n), P_* \leq O(d \log n/n)$, where the latter result is the first such result of its own.

*Instance 2 of QP.* Let us consider the following problem:

$$\min_{\mathbf{w} \in \mathcal{W}} P(\mathbf{w}) \triangleq \mathbb{E}_{\mathbf{z}}[\mathbf{w}^\top (S - \mathbf{z}\mathbf{z}^\top)\mathbf{w}] - \mathbf{w}^\top \mathbf{b} \tag{11}$$

where $S - \mathbb{E}_{\mathbf{z}}[\mathbf{z}\mathbf{z}^\top] \succeq 0$. It is notable that $f(\mathbf{w}, \mathbf{z}) = \mathbf{w}^\top (S - \mathbf{z}\mathbf{z}^\top)\mathbf{w} - \mathbf{w}^\top \mathbf{b}$ might be non-convex. A similar problem as (11) could arise in computing the leading eigen-vector of $\mathbb{E}[\mathbf{z}\mathbf{z}^\top]$ by performing shifted-and-inverted power method over random samples $\mathbf{z} \sim \mathbb{P}$ [10].

**Piecewise Linear Problems (PLP):** $\quad \min_{\mathbf{w} \in \mathcal{W}} P(\mathbf{w}) \triangleq \mathbb{E}[f(\mathbf{w}, \mathbf{z})] \tag{12}$

where $\mathbb{E}[f(\mathbf{w}, \mathbf{z})]$ is a piecewise linear convex function and $\mathcal{W}$ is a bounded polyhedron. We have the following corollary.

**Corollary 2.** *If $\mathbb{E}[f(\mathbf{w}, \mathbf{z})]$ is piecewise linear and convex and $\mathcal{W}$ is a bounded polyhedron, then the problem (12) satisfies $EBC(\theta = 1, \alpha)$. If $f(\mathbf{w}, \mathbf{z})$ is Lipschitz continuous, then ERM has a fast rate at least $\widetilde{O}(d/n)$, and ASA has a convergence rate of $\widetilde{O}(1/n)$. If $f(\mathbf{w}, \mathbf{z})$ is further non-negative and linear, then ERM has a fast rate of $\widetilde{O}((d/n)^2 + dP_*/n)$ when $n \geq \Omega(d \log n)$.*

*Instance 1 of PLP: minimizing the expected hinge loss for bounded data.* Consider the following problem:

$$\min_{\|\mathbf{w}\|_p \leq B} P(\mathbf{w}) \triangleq \mathbb{E}_{\mathbf{x}, y}[(1 - y\mathbf{w}^\top \mathbf{x})_+] \tag{13}$$

where $p = 1, \infty$ and $y \in \{1, -1\}$. Suppose that $\mathbf{x} \in \mathcal{X}$ is bounded and scaled such that $|\mathbf{w}^\top \mathbf{x}| \leq 1$. Koolen et al. [20] has considered this instance with $p = 2$ and proved that the Bernstein condition (Definition 2) holds with $\beta = 1$ for the problem (13) when $\mathbb{E}[y\mathbf{x}] \neq 0$ and $|\mathbf{w}^\top \mathbf{x}| \leq 1$. In contrast, we can show that the problem (13) with any $p = 1, 2, \infty$ norm constraint [3], the $EBC(\theta = 1, \alpha)$ holds since the objective $P(\mathbf{w}) = 1 - \mathbf{w}^\top \mathbb{E}[y\mathbf{x}]$ is essentially a linear function of $\mathbf{w}$. Then all results in Corollary 2 hold. To the best of our knowledge, the fast rates of ERM and SA for this instance with $\ell_1$ and $\ell_\infty$ norm constraint are the new results. In comparison, Koolen et al.'s [20] fast rate of $\widetilde{O}(1/n)$ only applies to SA and $\ell_2$ norm constraint, and their SA algorithm is not as efficient as our SA algorithm.

*Instance 2 of PLP: multi-dimensional newsvendor problem.* Consider a firm that manufactures $p$ products from $q$ resources. Suppose that a manager must decide on a resource vector $\mathbf{x} \in \mathbb{R}_+^q$ before the product demand vector $\mathbf{z} \in \mathbb{R}^p$ is observed. After the demand becomes known, the manager chooses a production vector $\mathbf{y} \in \mathbb{R}^p$ so as to maximize the operating profit. Assuming that the demand $\mathbf{z}$ is a random vector with discrete probability distribution, the problem is equivalent to

$$\min_{\mathbf{x} \in \mathbb{R}_+^q, \mathbf{x} \leq \mathbf{b}} \mathbf{c}^\top \mathbf{x} - \mathbb{E}[\Pi(\mathbf{x}; \mathbf{z})]$$

where both $\Pi(\mathbf{x}; \mathbf{z})$ and $\mathbb{E}[\Pi(\mathbf{x}; \mathbf{z})]$ are piecewise linear concave functions [18]. Then the problem fits to the setting in Corollary 2.

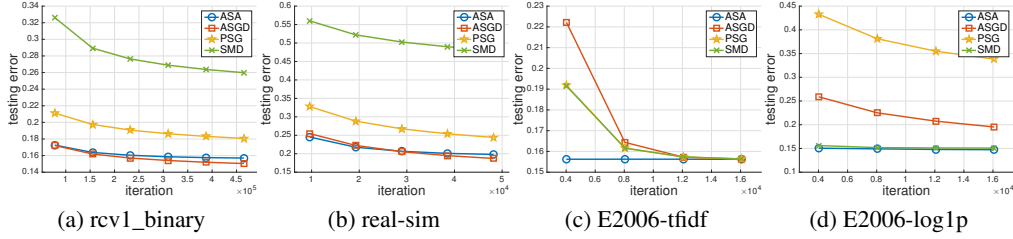

<center>(a) rcv1_binary      (b) real-sim      (c) E2006-tfidf      (d) E2006-log1p</center>

<center>Figure 1: Testing Error vs Iteration of ASA and other baselines for SA</center>

**Risk Minimization Problems over an $\ell_2$ ball.** Consider the following problem

$$\min_{\|\mathbf{w}\|_2 \leq B} P(\mathbf{w}) \triangleq \mathbb{E}_{\mathbf{z}}[f(\mathbf{w}, \mathbf{z})] \tag{14}$$

Assuming that $P(\mathbf{w})$ is convex and $\min_{\mathbf{w}\in\mathbb{R}^d} P(\mathbf{w}) < \min_{\|\mathbf{w}\|_2 \leq B} P(\mathbf{w})$, we can show that EBC$(\theta = 1, \alpha)$ holds (see supplement). Using this result, we can easily show that the considered problem (13) with $p = 2$ satisfies EBC$(\theta = 1, \alpha)$.

**Risk Minimization with $\ell_1$ Regularization Problems.** For $\ell_1$ regularized risk minimization:

$$\min_{\|\mathbf{w}\|_1 \leq B} P(\mathbf{w}) \triangleq \mathbb{E}[f(\mathbf{w}; \mathbf{z})] + \lambda\|\mathbf{w}\|_1, \tag{15}$$

we have the following corollary.

**Corollary 3.** *If the first component is quadratic as in (9) or is piecewise linear and convex, then the problem (15) satisfies EBC$(\theta = 1, \alpha)$. If the random function is Lipschitz continuous, then ERM has a fast rate at least $\widetilde{O}(d/n)$, and ASA has a convergence rate of $\widetilde{O}(1/n)$. If $f(\mathbf{w}, \mathbf{z})$ is further non-negative, convex and smooth, then ERM has a fast rate of $\widetilde{O}((d/n)^2 + dP_*/n)$ when $n \geq \Omega(d\log n)$.*

To the best of our knowledge, this above general result is the first of its kind. Next, we show some instances satisfying EBC$(\theta, \alpha)$ with $\theta < 1$. Consider the problem $\min_{\mathbf{w}\in\mathcal{W}} F(\mathbf{w}) \triangleq P(\mathbf{w}) + \lambda\|\mathbf{w}\|_p^p$, where $P(\mathbf{w})$ is quadratic as in (9), and $\mathcal{W}$ is a bounded polyhedron. In the supplement, we prove that EBC$(\theta = 2/p, \alpha)$ holds.

**A Case Study for ASA.** Finally, we provide some empirical evidence to support the effectiveness of the proposed ASA algorithm. In particular, we will consider solving an $\ell_1$ regularized expected square loss minimization problem (15) for learning a predictive model. We compare with two baselines whose convergence rate are known as $O(1/\sqrt{n})$, namely proximal stochastic gradient (PSG) method [7], and stochastic mirror descent (SMD) method using a $p$-norm divergence function ($p = 2\log d$) other than the Euclidean function. For SMD, we implement the algorithm proposed in [37], which was proposed for solving (15) and could be effective for very high-dimensional data. For ASA, we implement two versions that use PSG and SMD as the subroutine and report the one that gives the best performance. The two versions differ in using the Euclidean norm or the $p$-norm for measuring distance. Since the comparison is focused on the testing error, we also include another strong baseline, i.e, averaged stochastic gradient (ASGD) with a constant step size, which enjoys an $O(d/n)$ rate for minimizing the expected square loss without any constraints or regularizations [1].

We use four benchmark datasets from libsvm website[4], namely, real-sim, rcv1_binary, E2006-tfidf, E2006-log1p, whose dimensionality is 20958, 47236, 150360, 4272227, respectively. We divide each dataset into three sets, respectively training, validation, and testing. For E2006-tfidf and E2006-log1p dataset, we randomly split the given testing set into half validation and half testing. For the dataset real-sim which do not explicitly provides a testing set, we randomly split the entire data into 4:1:1 for training, validation, and testing. For rcv1_binary, despite that the test set is given, the size of the training set is relatively small. Thus we first combine the training and the testing sets and then follow the above procedure to split it.

The involved parameters of each algorithm are tuned based on the validation data. With the selected parameters, we run each algorithm by passing through training examples once and evaluate intermediate models on the testing data to compute the testing error measured by square loss. The results

on different data sets averaged over 5 random runs over shuffled training examples are shown in Figure 1. From the testing curves, we can see that the proposed ASA has similar convergence rate to ASGD on two relatively low-dimensional data sets. This is not surprise since both algorithms enjoy an $\widetilde{O}(1/n)$ convergence rate indicated by their theories. For the data set E2006-tfidf and E2006-log1p, we observe that ASA converges much faster than ASGD, which is due to the presence of $\ell_1$ regularization. In addition, ASA converges much faster than SGD and SMD with one exception on E2006-log1p, on which ASA performs slightly better than SMD.

## 6   Conclusion

We have comprehensively studied statistical learning under the error bound condition for both ERM and SA. We established the connection between the error bound condition and previous conditions for developing fast rates of empirical risk minimization for Lipschitz continuous loss functions. We also developed improved rates for non-negative and smooth convex loss functions, which induce faster rates that were not achieved before. Finally, we analyzed an efficient "parameter"-free SA algorithm under the error bound condition and showed that it is automatically adaptive to the error bound condition. Applications in machine learning and other fields are considered and empirical studies corroborate the fast rate of the developed algorithms. An open question is how to develop efficient SA algorithms under the error bound condition with optimistic rates for non-negative smooth loss functions similar to the results obtained for empirical risk minimization in this paper.

## Acknowledgement

The authors thank the anonymous reviewers for their helpful comments. M. Liu and T. Yang are partially supported by National Science Foundation (IIS-1545995). L. Zhang is partially supported by YESS (2017QNRC001). We thank Nishant A. Mehta for pointing out the work [12] for the proof of Theorem 1.

## Footnotes

[1] $\widetilde{O}$ hides a poly-logarithmic factor of $n$.

[2] One may consider $\theta \in (1, 2]$, which will yield the same order of excess risk bound as $\theta = 1$ in our settings.

[3]The case of $p = 2$ is showed later.

[4]http://www.csie.ntu.edu.tw/ cjlin/libsvmtools/datasets/

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
