[Supplementary Material · supplement_error_bound.pdf]

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

# A  Proof of Lemma 1

*Proof.* The proof follows similarly as the proof of Theorem 5.4 in [46]. Let us fix an arbitrary $\mathbf{w} \in \mathcal{W}$ and its closest optimal solution $\mathbf{w}^* \in \mathcal{W}_*$. Let $X = f(\mathbf{w}, \mathbf{z}) - f(\mathbf{w}^*, \mathbf{z})$ be a random variable due to $\mathbf{z}$. Then $|X| \leq 2GR \triangleq a$. Let $b > 0$ be any finite constant, $\kappa(x) = (e^x - x - 1)/x^2$ for $x \neq 0$ and $\kappa(0) = 1/2$, $c_1^b = 1/\kappa(2ba)$. Let $B = \alpha G^2$ and $v(x) = \frac{c_1^b}{B} x^{1-\theta} \wedge b$. Let $\varepsilon \geq 0$ and set $\eta = v(\varepsilon) \leq \frac{c_1^b}{B} \varepsilon^{1-\theta}$.

According to our analysis in the paper, we have established a similar condition to the Bernstein condition under our conditions, i.e.,
$$\mathbb{E}_{\mathbf{z}}[(f(\mathbf{w}, \mathbf{z}) - f(\mathbf{w}^*, \mathbf{z}))^2] \leq B(\mathbb{E}_{\mathbf{z}}[f(\mathbf{w}, \mathbf{z}) - f(\mathbf{w}^*, \mathbf{z})])^\theta$$
where $B = \alpha G^2$. Then
$$\mathrm{Var}[(f(\mathbf{w}, \mathbf{z}) - f(\mathbf{w}^*, \mathbf{z}))] \leq B(\mathbb{E}_{\mathbf{z}}[f(\mathbf{w}, \mathbf{z}) - f(\mathbf{w}^*, \mathbf{z})])^\theta$$

First, when $\varepsilon = 0$ we have $\eta = 0$, the $\mathbb{E}[e^{-\eta X}] \leq e^{\eta \varepsilon}$ hold trivially. Thus we focus on the case $\varepsilon > 0$, which implies that $\eta > 0$. Then Lemma 5.6 in [46] applied to the random variable $\eta$ gives
$$\mathbb{E}[X] + \frac{1}{\eta} \log \mathbb{E}[e^{-\eta X}] \leq \kappa(2ba) \eta \mathrm{Var}(X) \leq \kappa(2ba) \eta B(\mathbb{E}[X])^\theta \leq \varepsilon^{1-\theta}(\mathbb{E}[X])^\theta.$$

If $\varepsilon \leq \mathbb{E}[X]$, then $\varepsilon^{1-\theta}(\mathbb{E}[X])^\theta \leq \mathbb{E}[X]$, which implies $\frac{1}{\eta} \log \mathbb{E}[e^{-\eta X}] \leq 0 \leq \varepsilon$. This establishes the second part and the first part for $\varepsilon \leq \mathbb{E}[X]$. For $\varepsilon \geq \mathbb{E}[X]$, we have $\varepsilon^{1-\theta}(\mathbb{E}[X])^\theta \leq \varepsilon$. Then due to $\mathbb{E}[X] \geq 0$, we have $\frac{1}{\eta} \log \mathbb{E}[e^{-\eta X}] \leq \varepsilon$. $\qquad\square$

# B  Proof of Theorem 1

*Proof.* Let $F_{\mathbf{w}}(\mathbf{z}) = f(\mathbf{w}, \mathbf{z}) - f(\mathbf{w}^*, \mathbf{z})$, where $\mathbf{w}^*$ is the closest optimal solution to $\mathbf{w}$. Denote by $B = 2GR$. It is clear that $F_{\mathbf{w}}(\mathbf{z}) \leq B$. The goal is to show that with high probability, ERM does not select any $\mathbf{w} \in \mathcal{W}$ whose excess risk $P(\mathbf{w}) - P_* = \mathbb{E}_{\mathbf{z}}[F_{\mathbf{w}}(\mathbf{z})]$ is large than $\left(\frac{a}{n}\right)^{\frac{1}{2-\theta}}$ for some constant $a$. Clearly, with probability 1 ERM will never select any $\mathbf{w}$ for which both $F_{\mathbf{w}}(\mathbf{z}) > 0$ almost surely and with some positive probability $F_{\mathbf{w}}(\mathbf{z}) > 0$. These predictors are called the empirically inadmissible models. For any $\gamma_n > 0$, let $\mathcal{W}_{\geq \gamma_n}$ denote the subclass of models by starting with $\mathcal{W}$, retaining only models whose excess risk is at least $\gamma_n$, and further removing the empirically inadmissible models.

The goal now can be expressed equivalently as showing that, with high probability, ERM does not select any model $\mathbf{w} \in \mathcal{W}_{\geq \gamma_n}$, where $\gamma_n = \left(\frac{a}{n}\right)^{\frac{1}{2-\theta}}$. Let $\mathcal{W}_{\geq \gamma_n, \varepsilon}$ be the optimal proper $(\varepsilon/(2G))$-cover of $\mathcal{W}_{\geq \gamma_n}$. Note that this cover induces an $\varepsilon$-cover in sup norm over the function class $\{F_{\mathbf{w}} : \mathbf{w} \in \mathcal{W}_{\geq \gamma_n}\}$. To see this, for any $\mathbf{w} \in \mathcal{W}_{\geq \gamma_n}$, there exists $\widetilde{\mathbf{w}} \in \mathcal{W}_{\geq \gamma_n, \varepsilon}$ such that $\|\mathbf{w} - \widetilde{\mathbf{w}}\|_2 \leq \varepsilon/(2G)$. As a result,
$$\sup_{\mathbf{z}} |F_{\mathbf{w}}(\mathbf{z}) - F_{\widetilde{\mathbf{w}}}(\mathbf{z})| = \sup_{\mathbf{z}} |f(\mathbf{w}, \mathbf{z}) - f(\widetilde{\mathbf{w}}, \mathbf{z})| + \sup_{\mathbf{z}} |f(\mathbf{w}^*, \mathbf{z}) - f(\widetilde{\mathbf{w}}^*, \mathbf{z})|$$
$$\leq G\|\mathbf{w} - \widetilde{\mathbf{w}}\|_2 + G\|\mathbf{w}^* - \widetilde{\mathbf{w}}^*\|_2 \leq 2G\|\mathbf{w} - \widetilde{\mathbf{w}}\|_2 \leq \varepsilon,$$
where $\mathbf{w}^*, \widetilde{\mathbf{w}}^*$ are projections of $\mathbf{w}$ and $\widetilde{\mathbf{w}}$ onto $\mathcal{W}_*$ and the last inequality uses the non-expansiveness of the projection onto $\mathcal{W}_*$, which is convex due to the convexity of $P(\mathbf{w})$ and $\mathcal{W}$. Observe that the $\epsilon$-cover of $\mathcal{W}_{\geq \gamma_n} \subseteq \mathcal{B}^d(R)$ has cardinality at most $\left(\frac{4R}{\varepsilon}\right)^d$, and the cardinality of an optimal proper $\varepsilon$-cover is at most the cardinality of an optimal $\varepsilon/2$-cover. It hence follows that $|\mathcal{W}_{\geq \gamma_n, \varepsilon}| \leq \left(\frac{16GR}{\varepsilon}\right)^d$.

Let us consider a fixed $\mathbf{w} \in \mathcal{W}_{\geq \gamma_n, \varepsilon}$ and its closest optimal solution $\mathbf{w}^* \in \mathcal{W}_*$. According to Lemma 1, we have
$$\mathbb{E}_{\mathbf{z}}[e^{-v(\gamma_n)F_{\mathbf{w}}(\mathbf{z})}] \leq 1$$
Then using Theorem 13 in [12], where we set $u = B$ and $c = 1$, for all $\eta \in (0, v(\gamma_n))$ we have
$$\gamma_n \leq \mathbb{E}_{\mathbf{z}}[F_{\mathbf{w}}(\mathbf{z})] \leq -\frac{\eta B + 1}{1 - \eta/v(\gamma_n)} \frac{1}{\eta} \log \mathbb{E}_{\mathbf{z}}[e^{-\eta F_{\mathbf{w}}(\mathbf{z})}]$$
Let $\eta = v(\gamma_n)/2$, we have
$$\log \mathbb{E}_{\mathbf{z}}[e^{-(v(\gamma_n)/2)F_{\mathbf{w}}(\mathbf{z})}] \leq -\frac{0.5v(\gamma_n)}{Bv(\gamma_n) + 2} \gamma_n$$

Applying Theorem 1 in [27] with $t = \frac{\gamma_n}{2}$, we have

$$\Pr\left(\frac{1}{n}\sum_{i=1}^{n} F_{\mathbf{w}}(\mathbf{z}_i) \leq \frac{\gamma_n}{2}\right) \leq \exp\left(-\frac{0.5v(\gamma_n)}{Bv(\gamma_n)+2}n\gamma_n + \frac{v(\gamma_n)\gamma_n}{4}\right).$$

Assume that $\left(\frac{a}{n}\right)^{\frac{1-\theta}{2-\theta}} \leq \alpha b G^2 \kappa(4GRb)$, i.e., $n \geq a\left(\alpha b G^2 \kappa(4GRb)\right)^{(2-\theta)/(1-\theta)}$, which implies that $v(\gamma_n) = c\left(\frac{a}{n}\right)^{\frac{1-\theta}{2-\theta}} \wedge b = c\left(\frac{a}{n}\right)^{\frac{1-\theta}{2-\theta}}$ by noting the value of $c = 1/(\alpha G^2 \kappa(4GRb))$ in Lemma 1. Further we assume $n \geq a(0.5Bc)^{\frac{2-\theta}{1-\theta}}$. Hence $Bv(\gamma_n) \leq 2$.

$$\Pr\left(\frac{1}{n}\sum_{i=1}^{n} F_{\mathbf{w}}(\mathbf{z}_i) \leq \frac{\gamma_n}{2}\right) \leq \exp\left(-\frac{0.5v(\gamma_n)}{Bv(\gamma_n)+2}n\gamma_n + \frac{v(\gamma_n)\gamma_n}{4}\right) \leq \exp\left(-0.125v(\gamma_n)n\gamma_n + \frac{v(\gamma_n)\gamma_n}{4}\right)$$

$$= \exp\left(-0.125ca + \frac{ca}{4n}\right) \leq \exp\left(-0.375ca\right),$$

where we use $n \geq 1$.

As a result, we have

$$\Pr\left(\frac{1}{n}\sum_{i=1}^{n} F_{\mathbf{w}}(\mathbf{z}_i) \leq \frac{\gamma_n}{2}\right) \leq \exp\left(-0.375ca\right).$$

Taking a union bound over $\mathcal{W}_{\geq \gamma_n, \varepsilon}$ we have that

$$\Pr\left(\exists \mathbf{w} \in \mathcal{W}_{\geq \gamma_n, \varepsilon}, \frac{1}{n}\sum_{i=1}^{n} F_{\mathbf{w}}(\mathbf{z}_i) \leq \frac{\gamma_n}{2}\right) \leq \left(\frac{16GR}{\varepsilon}\right)^d \exp\left(-0.375ca\right)$$

Taking $\varepsilon = \frac{1}{2n^{1/(2-\theta)}}$ and $a = \frac{3}{c}(d\log(32GRn^{1/(2-\theta)}) + \log(1/\delta))$, with probability $1 - \delta$ for all $\mathbf{w} \in \mathcal{W}_{\geq \gamma_n, \varepsilon}$, we have $\frac{1}{n}\sum_{i=1}^{n} F_{\mathbf{w}}(\mathbf{z}_i) \geq \frac{a^{1/(2-\theta)}}{2n^{1/(2-\theta)}}$.

Now, since $\sup_{\mathbf{w} \in \mathcal{W}_{\geq \gamma_n}} \min_{\widetilde{\mathbf{w}} \in \mathcal{W}_{\geq \gamma_n, \varepsilon}} \|F_{\mathbf{w}} - F_{\widetilde{\mathbf{w}}}\|_\infty \leq \varepsilon = \frac{1}{2n^{1/(2-\theta)}}$, and by increasing $a$ by 1 to guarantee that $a > 1$, with probability $1 - \delta$, for all $\mathbf{w} \in \mathcal{W}_{\geq \gamma_n}$, $\frac{1}{n}\sum_{i=1}^{n} F_{\mathbf{w}}(\mathbf{z}_i) > 0$. $\qquad\square$

## C  Proof of Theorem 2

The proof follows the framework developed in [55], which converts the excess risk bound of $\widehat{\mathbf{w}}$ into large deviation of gradients. In particular, if we let $F(\mathbf{w}) = \mathbb{E}[f(\mathbf{w}; \mathbf{z})]$ and $F_n(\mathbf{w}) = \frac{1}{n}\sum_{i=1}^{n} f(\mathbf{w}; \mathbf{z}_i)$, we will prove the following lemma.

**Lemma 2.** *If we let $\widehat{\mathbf{w}}^*$ be an optimal solution to $\min_{\mathbf{w} \in \mathcal{W}} P(\mathbf{w})$ that is closest to $\widehat{\mathbf{w}}$, then we have*

$$P(\widehat{\mathbf{w}}) - P(\widehat{\mathbf{w}}^*) \leq \|\nabla F(\widehat{\mathbf{w}}) - \nabla F(\widehat{\mathbf{w}}^*) - [\nabla F_n(\widehat{\mathbf{w}}) - \nabla F_n(\widehat{\mathbf{w}}^*)]\|_2 \|\widehat{\mathbf{w}} - \widehat{\mathbf{w}}^*\|_2$$
$$+ \|\nabla F(\widehat{\mathbf{w}}^*) - \nabla F_n(\widehat{\mathbf{w}}^*)\|_2 \|\widehat{\mathbf{w}} - \widehat{\mathbf{w}}^*\|_2$$

*where $P(\mathbf{w}) = F(\mathbf{w}) + r(\mathbf{w})$.*

Note that [55] only proves the above result for $P(\mathbf{w}) = F(\mathbf{w})$. Then we use concentration inequalities, covering numbers, and a refined analysis leveraging the EBC to bound the excess risk, where the refined analysis leveraging the EBC is our main contribution for proving Theorem 2.

*Proof.* (Proof of Lemma 2)
$$P(\widehat{\mathbf{w}}) - P(\widehat{\mathbf{w}}^*) \leq \langle \partial P(\widehat{\mathbf{w}}), \widehat{\mathbf{w}} - \widehat{\mathbf{w}}^*\rangle = \langle \partial P(\widehat{\mathbf{w}}) - \partial P(\widehat{\mathbf{w}}^*), \widehat{\mathbf{w}} - \widehat{\mathbf{w}}^*\rangle + \langle \partial P(\widehat{\mathbf{w}}^*), \widehat{\mathbf{w}} - \widehat{\mathbf{w}}^*\rangle$$
$$= \langle \partial P(\widehat{\mathbf{w}}) - \partial P(\widehat{\mathbf{w}}^*) - [\partial P_n(\widehat{\mathbf{w}}) - \partial P_n(\widehat{\mathbf{w}}^*)], \widehat{\mathbf{w}} - \widehat{\mathbf{w}}^*\rangle + \langle \partial P_n(\widehat{\mathbf{w}}) - \partial P_n(\widehat{\mathbf{w}}^*) + \partial P(\widehat{\mathbf{w}}^*), \widehat{\mathbf{w}} - \widehat{\mathbf{w}}^*\rangle$$
$$= \langle \partial P(\widehat{\mathbf{w}}) - \partial P(\widehat{\mathbf{w}}^*) - [\partial P_n(\widehat{\mathbf{w}}) - \partial P_n(\widehat{\mathbf{w}}^*)], \widehat{\mathbf{w}} - \widehat{\mathbf{w}}^*\rangle + \langle \partial P(\widehat{\mathbf{w}}^*) - \partial P_n(\widehat{\mathbf{w}}^*), \widehat{\mathbf{w}} - \widehat{\mathbf{w}}^*\rangle$$
$$+ \langle \partial P_n(\widehat{\mathbf{w}}), \widehat{\mathbf{w}} - \widehat{\mathbf{w}}^*\rangle$$
According to the optimality condition of $\widehat{\mathbf{w}}$, there exists $\mathbf{v} \in \partial r(\widehat{\mathbf{w}})$ such that $\langle \nabla F_n(\widehat{\mathbf{w}}) + \mathbf{v}, \widehat{\mathbf{w}} - \widehat{\mathbf{w}}^*\rangle \leq 0$. Let $\partial P_n(\widehat{\mathbf{w}}) = \nabla F_n(\widehat{\mathbf{w}}) + \mathbf{v}$ and $\partial P(\widehat{\mathbf{w}}) = \nabla F(\widehat{\mathbf{w}}) + \mathbf{v}$ in the above inequality, we have
$$P(\widehat{\mathbf{w}}) - P(\widehat{\mathbf{w}}^*) \leq \langle \nabla F(\widehat{\mathbf{w}}) - \nabla F(\widehat{\mathbf{w}}^*) - [\nabla F_n(\widehat{\mathbf{w}}) - \nabla F_n(\widehat{\mathbf{w}}^*)], \widehat{\mathbf{w}} - \widehat{\mathbf{w}}^*\rangle$$
$$+ \langle \nabla F(\widehat{\mathbf{w}}^*) - \nabla F_n(\widehat{\mathbf{w}}^*), \widehat{\mathbf{w}} - \widehat{\mathbf{w}}^*\rangle$$
$$\leq (\|\nabla F(\widehat{\mathbf{w}}) - \nabla F(\widehat{\mathbf{w}}^*) - [\nabla F_n(\widehat{\mathbf{w}}) - \nabla F_n(\widehat{\mathbf{w}}^*)]\|_2 + \|\nabla F(\widehat{\mathbf{w}}^*) - \nabla F_n(\widehat{\mathbf{w}}^*)\|_2) \cdot \|\widehat{\mathbf{w}} - \widehat{\mathbf{w}}^*\|_2$$

$\square$

The proof below uses the $L$-smoothness and convexity of $F(\mathbf{w})$, i.e., there exists $L \geq 0$ such that for any $\mathbf{w}, \mathbf{u} \in \mathcal{W}$,

$$0 \leq f(\mathbf{w}, \mathbf{z}) - f(\mathbf{u}, \mathbf{z}) - \nabla f(\mathbf{u}, \mathbf{z})^\top (\mathbf{w} - \mathbf{u}) \leq \frac{L}{2} \|\mathbf{w} - \mathbf{u}\|_2^2, \quad \forall \mathbf{z} \in \mathcal{Z}.$$

We first prove the following theorem. Theorem 2 is a corollary of the following theorem by setting $\varepsilon = 1/n$.

**Theorem 4.** *Let $\varepsilon > 0$ be any constant and $C(\varepsilon) = 2(\log(2/\delta) + d\log(6R/\varepsilon)$. Under* **Assumptions 1, 2, 3***, and that $r(\mathbf{w})$ is convex and $G'$-Lipschitz continuous over $\mathcal{W}$, with probability at least $1 - 2\delta$, we have*

$$P(\widehat{\mathbf{w}}) - P_* \leq \frac{4(6LR^2 + \bar{G}R)C(\varepsilon)}{n} + 2\left(1 \vee \alpha^{1/\theta}\right)\left(\frac{4LC(\epsilon)P_*}{n}\right)^{\frac{1}{2-\theta}} + 2\left(12RL + \frac{\bar{G}}{4} + \frac{4LRC(\varepsilon)}{n}\right)\varepsilon,$$

*where $\bar{G} = G + G'$. Furthermore, if $n \geq \left(256LC(\varepsilon)\alpha^{\frac{1}{\theta}}\right)^{(2-\theta)}$, we also have*

$$P(\widehat{\mathbf{w}}) - P_* \leq 34LC(\varepsilon)\left(\frac{1}{n}\right)^{\frac{2}{2-\theta}} + 2\left(1 \vee 4\alpha^{1/\theta}\right)\left(\frac{\bar{G}C(\varepsilon)}{n}\right)^{\frac{2}{2-\theta}} + 2\left(1 \vee 64\alpha^{1/\theta}\right)\left(\frac{4LC(\varepsilon)P_*}{n}\right)^{\frac{1}{2-\theta}}$$

$$+ 4LC(\varepsilon)\left(1 \vee 64\alpha^{1/\theta}\right)\left(\frac{\varepsilon}{n}\right)^{\frac{2}{2-\theta}} + 12L\left(1 \vee 64\alpha^{1/\theta}\right)\varepsilon^{\frac{2}{2-\theta}} + 2\left(1 \vee 64\alpha^{1/\theta}\right)\left(\frac{4L\bar{G}C(\varepsilon)\varepsilon}{n}\right)^{\frac{1}{2-\theta}}.$$

To prove the theorem, we need the following lemmas.

**Lemma 3.** *Under* **Assumptions 1***, with probability at least $1 - \delta$, for any $\mathbf{w} \in \mathcal{W}$, we have*

$$\|\nabla F(\mathbf{w}) - \nabla F(\mathbf{w}^*) - [\nabla F_n(\mathbf{w}) - \nabla F_n(\mathbf{w}^*)]\|_2$$

$$\leq \frac{LC(\varepsilon)\|\mathbf{w} - \mathbf{w}^*\|_2}{n} + \frac{2LC(\varepsilon)\varepsilon}{n} + \sqrt{\frac{LC(\varepsilon)(P(\mathbf{w}) - P_*)}{n}} + 2\sqrt{\frac{L\bar{G}C(\varepsilon)\varepsilon}{n}} + 4L\varepsilon.$$

*where $\mathbf{w}^*$ is the closest optimal solution to $\mathbf{w}$ and $C(\varepsilon)$ is define in Theorem 4.*

**Lemma 4.** *Under* **Assumption 1***, with probability at least $1 - \delta$, for any $\mathbf{w}_* \in \mathcal{W}_*$, we have*

$$\|\nabla F(\mathbf{w}_*) - \nabla F_n(\mathbf{w}_*)\|_2 \leq \frac{GC(\varepsilon)}{n} + \sqrt{\frac{4LC(\varepsilon)P_*}{n}} + 2L\varepsilon. \tag{16}$$

**Lemma 5.** *Let $A$ be a nonnegative number. Under the $EBC(\theta, \alpha)$ condition with $\theta \in (0, 1]$ and $0 < \alpha < \infty$, for any $\epsilon > 0$ and $\mathbf{w} \in \mathcal{W}$, we have*

$$\|\mathbf{w} - \mathbf{w}^*\|_2 \sqrt{A} \leq \left(1 \vee \frac{\alpha^{1/\theta}}{4\epsilon}\right)A^{\frac{1}{2-\theta}} + \epsilon(P(\mathbf{w}) - P_*)$$

## C.1 Proof of Theorem 4

*Proof.* Using the Lemma 3 and Lemma 4 to proceed bounding the inequality in Lemma 2, with probability at least $1 - 2\delta$, we have

$$P(\widehat{\mathbf{w}}) - P_* \leq \frac{LC(\varepsilon)\|\widehat{\mathbf{w}} - \widehat{\mathbf{w}}^*\|_2^2}{n} + \frac{\bar{G}C(\varepsilon)\|\widehat{\mathbf{w}} - \widehat{\mathbf{w}}^*\|_2}{n} + \frac{2LC(\varepsilon)\varepsilon\|\widehat{\mathbf{w}} - \widehat{\mathbf{w}}^*\|_2}{n} + 6L\varepsilon\|\widehat{\mathbf{w}} - \widehat{\mathbf{w}}^*\|_2$$

$$+ \|\widehat{\mathbf{w}} - \widehat{\mathbf{w}}^*\|_2 \sqrt{\frac{LC(\varepsilon)(P(\widehat{\mathbf{w}}) - P_*)}{n}} + \|\widehat{\mathbf{w}} - \widehat{\mathbf{w}}^*\|_2 \sqrt{\frac{4LC(\varepsilon)P_*}{n}} + \|\widehat{\mathbf{w}} - \widehat{\mathbf{w}}^*\|_2 \sqrt{\frac{4L\bar{G}C(\varepsilon)\varepsilon}{n}}. \tag{17}$$

Next, we will bound the three terms that have a $1/\sqrt{n}$ factor.

$$\|\widehat{\mathbf{w}} - \widehat{\mathbf{w}}^*\|_2 \sqrt{\frac{LC(\varepsilon)(P(\widehat{\mathbf{w}}) - P_*)}{n}} \leq \frac{LC(\varepsilon)\|\widehat{\mathbf{w}} - \widehat{\mathbf{w}}^*\|_2^2}{n} + \frac{P(\widehat{\mathbf{w}}) - P_*}{4}, \tag{18}$$

$$\|\widehat{\mathbf{w}} - \widehat{\mathbf{w}}^*\|_2 \sqrt{\frac{4L\bar{G}C(\varepsilon)\varepsilon}{n}} \leq \frac{LC(\varepsilon)\|\widehat{\mathbf{w}} - \widehat{\mathbf{w}}^*\|_2^2}{n} + \bar{G}\varepsilon \tag{19}$$

$$\|\widehat{\mathbf{w}} - \widehat{\mathbf{w}}^*\|_2 \sqrt{\frac{4LC(\varepsilon)P_*}{n}} \leq \left(1 \vee \alpha^{1/\theta}\right) \left(\frac{4LC(\varepsilon)P_*}{n}\right)^{\frac{1}{2-\theta}} + \frac{P(\widehat{\mathbf{w}}) - P_*}{4} \qquad (20)$$

where the last inequality follows Lemma 5. Combining the inequalities in (17), (18), (19), and (20), with probability $1 - \delta$ we have

$$\frac{P(\widehat{\mathbf{w}}) - P_*}{2} \leq \frac{3LC(\varepsilon)\|\widehat{\mathbf{w}} - \widehat{\mathbf{w}}^*\|_2^2}{n} + \frac{\bar{G}C(\varepsilon)\|\widehat{\mathbf{w}} - \widehat{\mathbf{w}}^*\|_2}{n} + \frac{2LC(\varepsilon)\varepsilon\|\widehat{\mathbf{w}} - \widehat{\mathbf{w}}^*\|_2}{n} + 6L\varepsilon\|\widehat{\mathbf{w}} - \widehat{\mathbf{w}}^*\|_2$$

$$+ \bar{G}\varepsilon + \left(1 \vee \alpha^{1/\theta}\right)\left(\frac{4LC(\varepsilon)P_*}{n}\right)^{\frac{1}{2-\theta}}$$

$$\leq \frac{(12LR^2 + 2\bar{G}R)C(\varepsilon)}{n} + \left(1 \vee \alpha^{1/\theta}\right)\left(\frac{4LC(\varepsilon)P_*}{n}\right)^{\frac{1}{2-\theta}} + \left(12RL + \bar{G} + \frac{4LRC(\varepsilon)}{n}\right)\varepsilon,$$

which finishes the first part of the theorem.

To prove the second part, we need more refined analysis. The following inequalities will be proved later.

$$\frac{LC(\varepsilon)\|\widehat{\mathbf{w}} - \widehat{\mathbf{w}}^*\|_2^2}{n} \leq \max\left(LC(\varepsilon)\left(\frac{1}{n}\right)^{\frac{2}{2-\theta}}, \epsilon(P(\widehat{\mathbf{w}}) - P_*)\right), \text{ for } n \geq \left(LC(\varepsilon)\alpha^{\frac{1}{\theta}}/\epsilon\right)^{(2-\theta)} \qquad (21)$$

$$\|\widehat{\mathbf{w}} - \widehat{\mathbf{w}}^*\|_2 \sqrt{\frac{LC(\varepsilon)(P(\widehat{\mathbf{w}}) - P_*)}{n}} \leq \epsilon(P(\widehat{\mathbf{w}}) - P_*) + \frac{LC(\varepsilon)\|\widehat{\mathbf{w}} - \widehat{\mathbf{w}}^*\|_2^2}{4\epsilon n}$$

$$\leq \epsilon(P(\widehat{\mathbf{w}}) - P_*) + \max\left(\frac{LC(\varepsilon)}{\epsilon}\left(\frac{1}{n}\right)^{\frac{2}{2-\theta}}, \epsilon(P(\widehat{\mathbf{w}}) - P_*)\right), \text{ for } n \geq \left(LC(\varepsilon)\alpha^{\frac{1}{\theta}}/\epsilon^2\right)^{(2-\theta)} \qquad (22)$$

$$\frac{GC(\varepsilon)\|\widehat{\mathbf{w}} - \widehat{\mathbf{w}}^*\|_2}{n} \leq \left\{\left(1 \vee \frac{\alpha^{1/\theta}}{4\epsilon}\right)\left(\frac{GC(\varepsilon)}{n}\right)^{\frac{2}{2-\theta}} + \epsilon(P(\widehat{\mathbf{w}}) - P_*)\right\} \qquad (23)$$

$$\frac{2LC(\varepsilon)\varepsilon\|\widehat{\mathbf{w}} - \widehat{\mathbf{w}}^*\|_2}{n} \leq 2LC(\varepsilon)\left\{\left(1 \vee \frac{\alpha^{1/\theta}}{4\epsilon}\right)\left(\frac{\varepsilon}{n}\right)^{\frac{2}{2-\theta}} + \epsilon(P(\widehat{\mathbf{w}}) - P_*)\right\} \qquad (24)$$

$$6L\varepsilon\|\widehat{\mathbf{w}} - \widehat{\mathbf{w}}^*\|_2 \leq 6L\left\{\left(1 \vee \frac{\alpha^{1/\theta}}{4\epsilon}\right)\varepsilon^{\frac{2}{2-\theta}} + \epsilon(P(\widehat{\mathbf{w}}) - P_*)\right\} \qquad (25)$$

$$\|\widehat{\mathbf{w}} - \widehat{\mathbf{w}}^*\|_2 \sqrt{\frac{4LC(\varepsilon)P_*}{n}} \leq \left(1 \vee \frac{\alpha^{1/\theta}}{4\epsilon}\right)\left(\frac{4LC(\varepsilon)P_*}{n}\right)^{\frac{1}{2-\theta}} + \epsilon(P(\widehat{\mathbf{w}}) - P_*) \qquad (26)$$

$$\|\widehat{\mathbf{w}} - \widehat{\mathbf{w}}^*\|_2 \sqrt{\frac{4LGC(\varepsilon)\varepsilon}{n}} \leq \left(1 \vee \frac{\alpha^{1/\theta}}{4\epsilon}\right)\left(\frac{4LGC(\varepsilon)\varepsilon}{n}\right)^{\frac{1}{2-\theta}} + \epsilon(P(\widehat{\mathbf{w}}) - P_*) \qquad (27)$$

Plugging appropriate small constant values of $\epsilon$ in each inequality, we have

$$\frac{P(\widehat{\mathbf{w}}) - P_*}{2} \leq 17LC(\varepsilon)\left(\frac{1}{n}\right)^{\frac{2}{2-\theta}} + \left(1 \vee 4\alpha^{1/\theta}\right)\left(\frac{GC(\varepsilon)}{n}\right)^{\frac{2}{2-\theta}} + 2LC(\varepsilon)\left(1 \vee 64\alpha^{1/\theta}\right)\left(\frac{\varepsilon}{n}\right)^{\frac{2}{2-\theta}}$$

$$+ 6L\left(1 \vee 64\alpha^{1/\theta}\right)\varepsilon^{\frac{2}{2-\theta}} + \left(1 \vee 64\alpha^{1/\theta}\right)\left(\frac{4LGC(\varepsilon)\varepsilon}{n}\right)^{\frac{1}{2-\theta}} + \left(1 \vee \frac{\alpha^{1/\theta}}{4\epsilon}\right)\left(\frac{4LC(\varepsilon)P_*}{n}\right)^{\frac{1}{2-\theta}}.$$

$$\square$$

## C.2 Proof of Inequality (21)

*Proof.* If $\|\widehat{\mathbf{w}} - \widehat{\mathbf{w}}^*\|_2^2 \le (\frac{1}{n})^{\frac{\theta}{2-\theta}}$, then $\frac{LC(\varepsilon)\|\widehat{\mathbf{w}} - \widehat{\mathbf{w}}^*\|_2^2}{n} \le LC(\varepsilon)(\frac{1}{n})^{\frac{2}{2-\theta}}$. If $\|\widehat{\mathbf{w}} - \widehat{\mathbf{w}}^*\|_2^2 \ge (\frac{1}{n})^{\frac{\theta}{2-\theta}}$, then

$$\frac{1}{\|\widehat{\mathbf{w}} - \widehat{\mathbf{w}}^*\|_2^{\frac{2}{\theta} - 2}} \le n^{\frac{1-\theta}{2-\theta}}, \tag{28}$$

so when $n \ge \left(LC(\varepsilon)\alpha^{\frac{1}{\theta}}/\epsilon\right)^{(2-\theta)}$, we have

$$\frac{LC(\varepsilon)\|\widehat{\mathbf{w}} - \widehat{\mathbf{w}}^*\|_2^2}{n} = \frac{LC(\varepsilon)\|\widehat{\mathbf{w}} - \widehat{\mathbf{w}}^*\|_2^{\frac{2}{\theta}}\|\widehat{\mathbf{w}} - \widehat{\mathbf{w}}^*\|_2^{2 - \frac{2}{\theta}}}{n} \le \frac{LC(\varepsilon)\alpha^{\frac{1}{\theta}}(P(\widehat{\mathbf{w}}) - P_*)}{n^{\frac{1}{2-\theta}}} \le \epsilon(P(\widehat{\mathbf{w}}) - P_*),$$

where the first inequality holds by employing the EBC and the inequality (28), and the second inequality holds due to the fact that $n \ge \left(LC(\varepsilon)\alpha^{\frac{1}{\theta}}/\epsilon\right)^{(2-\theta)}$. Combining two cases together, we complete the proof. $\qquad\square$

## C.3 Proof of Inequality (22)

*Proof.* The first inequality in the inequality (22) obviously holds, and now we prove the second inequality.

- If $\|\widehat{\mathbf{w}} - \widehat{\mathbf{w}}^*\|_2^2 \le 4(\frac{1}{n})^{\frac{\theta}{2-\theta}}$, then
$$\frac{LC(\varepsilon)\|\widehat{\mathbf{w}} - \widehat{\mathbf{w}}^*\|_2^2}{4\epsilon n} \le \frac{LC(\varepsilon)}{\epsilon}(\frac{1}{n})^{\frac{2}{2-\theta}}.$$

- If $\|\widehat{\mathbf{w}} - \widehat{\mathbf{w}}^*\|_2^2 \ge 4(\frac{1}{n})^{\frac{\theta}{2-\theta}}$, then
$$\frac{1}{\|\widehat{\mathbf{w}} - \widehat{\mathbf{w}}^*\|_2^{2 - \frac{2}{\theta}}} \ge \frac{1}{2^{2 - \frac{2}{\theta}}}n^{\frac{\theta-1}{2-\theta}} \ge \frac{1}{4}n^{\frac{\theta-1}{2-\theta}}, \tag{29}$$
so when $n \ge \left(LC(\varepsilon)\alpha^{\frac{1}{\theta}}/\epsilon^2\right)^{(2-\theta)}$, we have
$$\frac{LC(\varepsilon)\|\widehat{\mathbf{w}} - \widehat{\mathbf{w}}^*\|_2^2}{4\epsilon n} = \frac{LC(\varepsilon)\|\widehat{\mathbf{w}} - \widehat{\mathbf{w}}^*\|_2^{\frac{2}{\theta}}\|\widehat{\mathbf{w}} - \widehat{\mathbf{w}}_*\|_2^{2 - \frac{2}{\theta}}}{4\epsilon n} \le \frac{LC(\varepsilon)\alpha^{\frac{1}{\theta}}(P(\widehat{\mathbf{w}}) - P_*)4n^{\frac{1-\theta}{2-\theta}}}{4\epsilon n}$$
$$\le \epsilon(P(\widehat{\mathbf{w}}) - P_*),$$
where the first inequality holds by employing the EBC and the inequality (29), and the second inequality holds due to the fact that $n \ge \left(LC(\varepsilon)\alpha^{\frac{1}{\theta}}/\epsilon^2\right)^{(2-\theta)}$.

Combining two cases together, we complete the proof. $\qquad\square$

## C.4 Proof of Inequalities (23)–(27)

*Proof.* In Lemma 5, taking $A$ to be
$$\left(\frac{GC(\varepsilon)}{n}\right)^2, \left(\frac{\varepsilon}{n}\right)^2, \varepsilon^2, \frac{4LC(\varepsilon)P_*}{n}, \frac{4LGC(\varepsilon)\varepsilon}{n}$$
yields inequalities (23)–(27) respectively. $\qquad\square$

# D Proof of Lemma 3

**Lemma 6.** *[40]. Let $\mathcal{H}$ be a Hilbert space and let $\xi$ be a random variable with values in $\mathcal{H}$. Assume $\|\xi\| \le G < \infty$ almost surely. Denote $\sigma^2(\xi) = \mathbb{E}\left[\|\xi\|^2\right]$. Let $\{\xi_i\}_{i=1}^m$ be $m$ ($m < \infty$) independent drawers of $\xi$. For any $0 < \delta < 1$, with confidence $1 - \delta$,*
$$\left\|\frac{1}{m}\sum_{i=1}^m [\xi_i - \mathbb{E}[\xi_i]]\right\| \le \frac{2G\log(2/\delta)}{m} + \sqrt{\frac{2\sigma^2(\xi)\log(2/\delta)}{m}}.$$

*Proof of Lemma 3.* In order to prove the high probability bounds for all $\mathbf{w} \in \mathcal{W}$, we first consider the points in the $\varepsilon$-net of $\mathcal{W}$ with minimal cardinality. To this end, let $\mathcal{N}(\mathcal{W}, \varepsilon)$ denote the $\varepsilon$-net of $\mathcal{W}$ with minimal cardinality. Since $\mathcal{W} \subseteq \mathcal{B}^d(R)$, where $\mathcal{B}^d(R)$ denotes a $d$-dimensional bounded ball with radius $R$. Following the standard results of covering numbers, we have

$$\log |\mathcal{N}(\mathcal{W}, \varepsilon)| \le \log |\mathcal{N}(\mathcal{B}^d(R), \varepsilon/2)| \le d \log \frac{6R}{\epsilon}.$$

We first consider a fixed $\mathbf{w} \in \mathcal{N}(\mathcal{W}, \varepsilon)$. Denote by $\mathbf{w}^*$ the closest optimal solution to $\mathbf{w}$. Let $f_i(\mathbf{w}) = f(\mathbf{w}, \mathbf{z}_i)$. Since $f_i(\cdot)$ is $L$-smooth, we have

$$\|\nabla f_i(\mathbf{w}) - \nabla f_i(\mathbf{w}^*)\|_2 \le L \|\mathbf{w} - \mathbf{w}^*\|_2. \tag{30}$$

Because $f_i(\cdot)$ is both convex and $L$-smooth, by (2.1.7) of [30], we have

$$\|\nabla f_i(\mathbf{w}) - \nabla f_i(\mathbf{w}^*)\|_2^2 \le L \left( f_i(\mathbf{w}) - f_i(\mathbf{w}^*) - \langle \nabla f_i(\mathbf{w}_*), \mathbf{w} - \mathbf{w}^* \rangle \right).$$

Taking expectation over both sides, we have

$$\mathbb{E} \left[ \|\nabla f_i(\mathbf{w}) - \nabla f_i(\mathbf{w}^*)\|_2^2 \right] \le L \left( F(\mathbf{w}) - F(\mathbf{w}^*) - \langle \nabla F(\mathbf{w}^*), \mathbf{w} - \mathbf{w}^* \rangle \right) \le L \left( P(\mathbf{w}) - P(\mathbf{w}^*) \right),$$

where the last inequality follows from the optimality condition of $\mathbf{w}^*$, i.e., there exists $\mathbf{v}_* \in \partial R(\mathbf{w}^*)$

$$\langle \nabla F(\mathbf{w}^*) + \mathbf{v}_*, \mathbf{w} - \mathbf{w}_* \rangle \ge 0, \ \forall \mathbf{w} \in \mathcal{W}.$$

and the convexity of $R(\mathbf{w})$ and $F(\mathbf{w})$, i.e., $\langle \nabla F(\mathbf{w}^*), \mathbf{w} - \mathbf{w}^* \rangle \le F(\mathbf{w}) - F(\mathbf{w}^*)$ and $\langle \mathbf{v}_*, \mathbf{w} - \mathbf{w}^* \rangle \le R(\mathbf{w}) - R(\mathbf{w}^*)$.

Following Lemma 6, with probability at least $1 - \delta$, we have

$$\left\| \nabla F(\mathbf{w}) - \nabla F(\mathbf{w}^*) - \frac{1}{n} \sum_{i=1}^{n} [\nabla f_i(\mathbf{w}) - \nabla f_i(\mathbf{w}^*)] \right\|_2 \le \frac{2L \|\mathbf{w} - \mathbf{w}^*\|_2 \log(2/\delta)}{n}$$

$$+ \sqrt{\frac{2L(P(\mathbf{w}) - P(\mathbf{w}^*)) \log(2/\delta)}{n}}.$$

By taking the union bound over $\mathcal{N}(\mathcal{W}, \varepsilon)$, we have for any $\mathbf{w} \in \mathcal{N}(\mathcal{W}, \varepsilon)$, with probability $1 - \delta$,

$$\|\nabla P(\mathbf{w}) - \nabla P(\mathbf{w}^*) - [\nabla P_n(\mathbf{w}) - \nabla P_n(\mathbf{w}^*)]\|_2 = \left\| \nabla F(\mathbf{w}) - \nabla F(\mathbf{w}^*) - \frac{1}{n} \sum_{i=1}^{n} [\nabla f_i(\mathbf{w}) - \nabla f_i(\mathbf{w}^*)] \right\|_2$$

$$\le \frac{2L \|\mathbf{w} - \mathbf{w}_*\|_2 (\log(2/\delta) + d \log(6R/\varepsilon))}{n} + \sqrt{\frac{2L(P(\mathbf{w}) - P(\mathbf{w}^*))(\log(2/\delta) + d \log(6R/\varepsilon))}{n}}.$$

To finish the proof of Lemma 3, for any $\mathbf{w} \in \mathcal{W}$. There exists $\widetilde{\mathbf{w}} \in \mathcal{N}(\mathcal{W}, \varepsilon)$ such that $\|\mathbf{w} - \widetilde{\mathbf{w}}\| \le \varepsilon$. Let $\widetilde{\mathbf{w}}^*$ denote the closest optimal solution to $\widetilde{\mathbf{w}}$. Then by non-expansiveness of projection onto a convex set we have $\|\mathbf{w}^* - \widetilde{\mathbf{w}}^*\|_2 \le \|\mathbf{w} - \widetilde{\mathbf{w}}\|_2 \le \varepsilon$. In addition, we have

$$\|\widetilde{\mathbf{w}} - \widetilde{\mathbf{w}}^*\|_2 \le \|\widetilde{\mathbf{w}} - \mathbf{w}\|_2 + \|\mathbf{w} - \mathbf{w}^*\|_2 + \|\mathbf{w}^* - \widetilde{\mathbf{w}}^*\|_2 \le 2\varepsilon + \|\mathbf{w} - \mathbf{w}^*\|_2 \tag{31}$$

$$P(\widetilde{\mathbf{w}}) - P(\widetilde{\mathbf{w}}^*) \le P(\widetilde{\mathbf{w}}) - P(\mathbf{w}) + P(\mathbf{w}) - P(\mathbf{w}^*) + P(\mathbf{w}^*) - P(\widetilde{\mathbf{w}}^*)$$
$$\le \bar{G} \|\widetilde{\mathbf{w}} - \mathbf{w}\|_2 + P(\mathbf{w}) - P(\mathbf{w}^*) + \bar{G} \|\mathbf{w}^* - \widetilde{\mathbf{w}}^*\|_2 \tag{32}$$
$$\le 2\bar{G}\varepsilon + P(\mathbf{w}) - P(\mathbf{w}^*)$$

Then with probability $1 - \delta$, we have

$$\|\nabla P(\mathbf{w}) - \nabla P(\mathbf{w}^*) - [\nabla P_n(\mathbf{w}) - \nabla P_n(\mathbf{w}^*)]\|_2$$

$$\le \|\nabla P(\widetilde{\mathbf{w}}) - \nabla P(\widetilde{\mathbf{w}}^*) - [\nabla P_n(\widetilde{\mathbf{w}}) - \nabla P_n(\widetilde{\mathbf{w}}^*)]\|_2 + 2L \|\mathbf{w} - \widetilde{\mathbf{w}}\|_2 + 2L \|\mathbf{w}^* - \widetilde{\mathbf{w}}^*\|_2$$

$$\le \frac{2L \|\widetilde{\mathbf{w}} - \widetilde{\mathbf{w}}^*\|_2 (\log(2/\delta) + 2d \log(6R/\varepsilon))}{n} + \sqrt{\frac{2L(P(\widetilde{\mathbf{w}}) - P(\widetilde{\mathbf{w}}^*))(\log(2/\delta) + 2d \log(6R/\varepsilon))}{n}} + 4L\varepsilon$$

$$\le \frac{2L(\|\mathbf{w} - \mathbf{w}^*\|_2 + 2\varepsilon)(\log(2/\delta) + 2d \log(6R/\varepsilon))}{n} + \sqrt{\frac{2L(2\bar{G}\varepsilon + (P(\mathbf{w}) - P(\mathbf{w}^*)))(\log(2/\delta) + 2d \log(6R/\varepsilon))}{n}} +$$

$4L\varepsilon$

$$\le \frac{LC(\varepsilon)\|\mathbf{w} - \mathbf{w}^*\|_2}{n} + \frac{2LC(\varepsilon)\varepsilon}{n} + \sqrt{\frac{LC(\varepsilon)(P(\mathbf{w}) - P_*)}{n}} + 2\sqrt{\frac{L\bar{G}C(\varepsilon)\varepsilon}{n}} + 4L\varepsilon.$$

$\square$

# E Proof of Lemma 4

*Proof.* We first consider a fixed $\mathbf{w}_* \in \mathcal{N}(\mathcal{W}_*, \varepsilon) \subseteq \mathcal{W}_*$. To apply Lemma 6, we need an upper bound of $\mathbb{E}\left[\|\nabla f_i(\mathbf{w}_*)\|_2^2\right]$. Since $f_i(\cdot)$ is $L$-smooth and nonnegative, from Lemma 4.1 of [41], we have

$$\|\nabla f_i(\mathbf{w}_*)\|_2^2 \le 4L f_i(\mathbf{w}_*)$$

and thus

$$\mathbb{E}\left[\|\nabla f_i(\mathbf{w}_*)\|_2^2\right] \le 4L\mathbb{E}\left[f_i(\mathbf{w}_*)\right] = 4LF_*.$$

By **Assumption 1**, we have $\|\nabla f_i(\mathbf{w}_*)\|_2 \le G$. Then, according to Lemma 6, with probability at least $1 - \delta$, we have

$$\|\nabla F(\mathbf{w}_*) - \nabla F_n(\mathbf{w}_*)\|_2 = \left\|\nabla F(\mathbf{w}_*) - \frac{1}{n}\sum_{i=1}^{n}\nabla f_i(\mathbf{w}_*)\right\|_2 \le \frac{2G\log(2/\delta)}{n} + \sqrt{\frac{8LF_*\log(2/\delta)}{n}}.$$

By taking the union bound over $\mathcal{N}(\mathcal{W}_*, \varepsilon)$, for any $\mathbf{w}_* \in \mathcal{N}(\mathcal{W}_*, \varepsilon)$, with probability $1 - \delta$ we have

$$\|\nabla F(\mathbf{w}_*) - \nabla F_n(\mathbf{w}_*)\|_2 \le \frac{GC(\varepsilon)}{n} + \sqrt{\frac{4LF_*C(\varepsilon)}{n}}.$$

For any $\mathbf{w}^* \in \mathcal{W}_*$, there exists $\widetilde{\mathbf{w}}^* \in \mathcal{N}(\mathcal{W}_*, \varepsilon)$ such that $\|\mathbf{w}^* - \widetilde{\mathbf{w}}^*\| \le \varepsilon$. Then

$$\|\nabla F(\mathbf{w}^*) - \nabla F_n(\mathbf{w}^*)\|_2 \le \|\nabla F(\widetilde{\mathbf{w}}^*) - \nabla F_n(\widetilde{\mathbf{w}}^*)\|_2 + \|\nabla F(\mathbf{w}^*) - \nabla F(\widetilde{\mathbf{w}}^*)\|_2$$
$$+ \|\nabla F_n(\mathbf{w}^*) - \nabla F_n(\widetilde{\mathbf{w}}^*)\|_2$$
$$\le \frac{GC(\varepsilon)}{n} + \sqrt{\frac{4LF_*C(\varepsilon)}{n}} + 2L\varepsilon.$$

$\square$

# F Proof of Lemma 5

*Proof.* We consider two cases. First, $\|\mathbf{w} - \mathbf{w}^*\|_2 \le A^{\frac{\theta}{4-2\theta}}$, under which the inequality follows trivially. Next, we consider $\|\mathbf{w} - \widehat{\mathbf{w}}^*\|_2 \ge A^{\frac{\theta}{4-2\theta}}$. Then

$$\|\mathbf{w} - \mathbf{w}^*\|_2 \sqrt{A} = \frac{\|\mathbf{w} - \mathbf{w}^*\|_2^{1/\theta}}{\|\mathbf{w} - \mathbf{w}^*\|_2^{1/\theta-1}}\sqrt{A} \le \|\mathbf{w} - \mathbf{w}^*\|_2^{1/\theta} A^{\frac{1}{2(2-\theta)}} \le \frac{\epsilon\|\mathbf{w} - \mathbf{w}^*\|_2^{2/\theta}}{\alpha^{1/\theta}} + \frac{\alpha^{1/\theta}}{4\epsilon}A^{\frac{1}{2-\theta}}$$

$$\le \epsilon(P(\mathbf{w}) - P_*) + \frac{\alpha^{1/\theta}}{4\epsilon}A^{\frac{1}{2-\theta}}$$

where the last inequality follows the EBC. $\square$

# G Proof of Theorem 3

Before proceeding to the proof, we first present a standard result for SSG, which is the Lemma 10 of [14].

**Proposition 1.** *Suppose* **Assumptions 1 and 2** *hold. Let* $0 < \delta < 1$, $\mathbf{w}^* \in \mathcal{W}_*$ *be the closest optimal solution to* $\mathbf{w}_1$, *and* $R_0$ *be an upper bound on* $\|\mathbf{w}_1 - \mathbf{w}^*\|_2$. *Apply* $T$ *iterations of the update* $\mathbf{w}_{t+1} = \Pi_{\mathcal{W}\cap\mathcal{B}(\mathbf{w}_1,R_0)}(\mathbf{w}_t - \gamma g_t)$, *where* $g_t$ *is a stochastic subgradient of* $P(\mathbf{w})$ *at* $\mathbf{w}_t$. *With probability at least* $1 - \delta$, *we have*

$$P(\widehat{\mathbf{w}}_T) - P_* \le \frac{\gamma G^2}{2} + \frac{\|\mathbf{w}_1 - \mathbf{w}^*\|_2^2}{2\gamma(T+1)} + \frac{4GR_0\sqrt{2\log(2/\delta)}}{\sqrt{T+1}}.$$

*where* $\widehat{\mathbf{w}}_T = \frac{1}{T+1}\sum_{t=1}^{T+1}\mathbf{w}_t$. *Moreover, choose* $\gamma = \frac{R_0}{G\sqrt{T+1}}$, *and then with probability at least* $1 - \delta$,

$$P(\widehat{\mathbf{w}}_T) - P_* \le R_0 G \left(\frac{1}{\sqrt{T+1}} + \frac{4\sqrt{2\log(2/\delta)}}{\sqrt{T+1}}\right).$$

It is easy to derive a similar lemma as Proposition 1, which is stated in Lemma 7.

**Lemma 7.** *Suppose* **Assumptions 1 and 2** *hold. Let* $0 < \delta < 1$, $R_0$ *be any nonnegative real number. Apply* $T$ *iterations of the update* $\mathbf{w}_{t+1} = \Pi_{\mathcal{W}\cap\mathcal{B}(\mathbf{w}_1,R_0)}(\mathbf{w}_t - \gamma g_t)$, *where* $g_t$ *is a stochastic*

*subgradient of $P(\mathbf{w})$ at $\mathbf{w}_t$. With probablity at least $1 - \delta$, we have*

$$P(\widehat{\mathbf{w}}_T) - P(\mathbf{w}_1) \leq \frac{\gamma G^2}{2} + \frac{4GR_0\sqrt{2\log(2/\delta)}}{\sqrt{T+1}},$$

*where $\widehat{\mathbf{w}}_T = \frac{1}{T+1}\sum_{t=1}^{T+1}\mathbf{w}_t$. Moreover, choose $\gamma = \frac{R_0}{G\sqrt{T+1}}$, and then with probability at least $1 - \delta$,*

$$P(\widehat{\mathbf{w}}_T) - P(\mathbf{w}_1) \leq R_0 G\left(\frac{1}{\sqrt{T+1}} + \frac{4\sqrt{2\log(2/\delta)}}{\sqrt{T+1}}\right).$$

*Proof.* Denote $\mathbb{E}_{t-1}(X)$ by the expectation conditioned on the randomness until round $t - 1$, then we have $\mathbb{E}_{t-1}(\hat{g}_t) = g_t$, and $X_t = g_t(\mathbf{w}_t - \mathbf{w}_1) - \hat{g}_t(\mathbf{w}_t - \mathbf{w}_1)$ is a martingale difference sequence. Note that $\|g_t\|_2 = \|\mathbb{E}_{t-1}(\hat{g}_t)\|_2 \leq \mathbb{E}_{t-1}(\|\hat{g}_t\|_2) \leq G$, so we have

$$|X_t| \leq \|g_t\|_2\|\mathbf{w}_t - \mathbf{w}_1\|_2 + \|\hat{g}_t\|_2\|\mathbf{w}_t - \mathbf{w}_1\|_2 \leq 4GR_0,$$

since the update needs to project the gradient update onto the intersection of $\mathcal{W}$ and a ball with radius $R_0$.

By Azuma-Hoeffding's inequality, we have with probability at least $1 - \delta$,

$$\frac{1}{T+1}\sum_{t=1}^{T+1} g_t(\mathbf{w}_t - \mathbf{w}_1) - \frac{1}{T+1}\sum_{t=1}^{T}\hat{g}_t(\mathbf{w}_t - \mathbf{w}_1) \leq \frac{4GR_0\sqrt{2\log(1/\delta)}}{\sqrt{T+1}}. \tag{33}$$

By the convexity of $P$, we have $P(\mathbf{w}_t) - P(\mathbf{w}_1) \leq g_t(\mathbf{w}_t - \mathbf{w}_1)$, then using a standard result in online gradient descent [56], we have

$$\frac{1}{T+1}\sum_{t=1}^{T}\hat{g}_t(\mathbf{w}_t - \mathbf{w}_1) \leq \frac{\gamma G^2}{2} + \frac{\|\mathbf{w}_1 - \mathbf{w}_1\|_2^2}{2\gamma(T+1)} = \frac{\gamma G^2}{2}. \tag{34}$$

Combining inequality (33) and (34) suffices to derive the conclusion. □

With the above proposition and lemma, the proof of Theorem 3 proceeds similarly as that of Theorem 5.3 in [15]. The difference is that our analysis only relies on the EBC instead of the uniform convexity.

*Proof.* Define $\bar{\delta} = \frac{2\delta}{\log_2 n}$, and

$$a(n, \bar{\delta}) = G\left(\frac{1}{\sqrt{n+1}} + \frac{4\sqrt{2\log(2/\bar{\delta})}}{\sqrt{n+1}}\right).$$

We set $\mu_0 = 2R_0^{1-\frac{2}{\theta}}a(n_0, \bar{\delta})$, $\mu_k = 2^{(\frac{2}{\theta}-1)k}\mu_0$ and $R_k = R_0/2^k$, where $k = 1, \ldots, m$. Then we have $\mu_k R_k^{\frac{2}{\theta}} = 2^{-k}\mu_0 R_0^{\frac{2}{\theta}}$. We can also assume that $\alpha$ is large enough such that $\alpha \geq R_0^{2-\theta}/G^\theta$, i.e., $\alpha^{-\frac{1}{\theta}} \leq GR_0^{1-\frac{2}{\theta}}$, otherwise we can set $\alpha = R_0^{2-\theta}/G^\theta$, which makes the EBC still hold.

By definition of $m$, when $n \geq 100$,

$$0 < \frac{1}{2}\log_2\frac{2n}{\log_2 n} - 2 \leq m \leq \frac{1}{2}\log_2\frac{2n}{\log_2 n} - 1 \leq \frac{1}{2}\log_2 n, \tag{35}$$

so we have

$$2^m \geq \frac{1}{4}\sqrt{\frac{2n}{\log_2 n}}. \tag{36}$$

When $n \geq 100$, we have

$$\mu_m = 2^{(\frac{2}{\theta}-1)m}\mu_0 \geq 2^m \mu_0$$

$$\geq \frac{1}{4}\sqrt{\frac{2n}{\log_2 n}}4GR_0^{1-\frac{2}{\theta}}\left(\frac{1}{2\sqrt{n_0+1}} + \frac{2\sqrt{2\log(\log_2 n)}}{\sqrt{n_0+1}}\right)$$

$$\geq GR_0^{1-\frac{2}{\theta}}\sqrt{\frac{2n}{\log_2 n}}\left(\frac{1}{2\sqrt{\frac{n}{m}+1}} + \frac{2\sqrt{2\log(\log_2 n)}}{\sqrt{\frac{n}{m}+1}}\right)$$

$$\geq GR_0^{1-\frac{2}{\theta}}\sqrt{\frac{2n}{\log_2 n}}\left(\frac{1}{2\sqrt{\frac{2n}{\log_2 2n - \log_2 \log_2 n - 4}+1}} + \frac{2\sqrt{2\log(\log_2 n)}}{\sqrt{\frac{2n}{\log_2 2n - \log_2 \log_2 n - 4}+1}}\right)$$

$$\geq GR_0^{1-\frac{2}{\theta}}\sqrt{\frac{2n}{\log_2 n}}\frac{2\sqrt{\sqrt{2\log(\log_2 n)}}}{\sqrt{\frac{2n}{\log_2 2n - \log_2 \log_2 n - 4}+1}}$$

$$= GR_0^{1-\frac{2}{\theta}}\frac{2\sqrt{\sqrt{2\log(\log_2 n)}}}{\sqrt{\frac{1}{1-\frac{\log_2 \log_2 n+3}{\log_2 n}} + \frac{\log_2 n}{2n}}} \geq GR_0^{1-\frac{2}{\theta}},$$

where the first inequality holds because $\theta \in (0, 1]$, the second inequality comes from (36) and the fact that $0 < \delta < 1$, the third and fourth inequalities hold because of the definition of $n_0$ and inequality (35), the fifth inequality holds by utilizing $a+b \geq 2\sqrt{ab}$, and the sixth inequality holds since $n \geq 100$ and the function is monotonically increasing with respect to $n$. So $\alpha^{-\frac{1}{\theta}} \leq \mu_m$.

Below, given $\widehat{\mathbf{w}}_k$ we denote by $\widehat{\mathbf{w}}_k^*$ the closest optimal solution to $\widehat{\mathbf{w}}_k$. Next, we consider two cases.

**Case 1.** If $\alpha^{-\frac{1}{\theta}} \geq \mu_0$, then $\mu_0 \leq \alpha^{-\frac{1}{\theta}} \leq \mu_m$. We have the following lemma.

**Lemma 8.** *Let $k^*$ satisfy $\mu_{k^*} \leq \alpha^{-\frac{1}{\theta}} \leq 2^{\frac{2}{\theta}-1}\mu_{k^*}$. Then for any $1 \leq k \leq k^*$, there exists a Borel set $\mathcal{A}_k \subset \Omega$ of probability at least $1 - k\bar{\delta}$, such that for $\omega \in \mathcal{A}_k$, the points $\{\widehat{\mathbf{w}}_k\}_{k=1}^m$ generated by the Algorithm 2 satisfy*

$$\|\widehat{\mathbf{w}}_{k-1} - \widehat{\mathbf{w}}_{k-1}^*\|_2 \leq R_{k-1} = 2^{-k+1}R_0, \tag{37}$$

$$P(\widehat{\mathbf{w}}_k) - P_* \leq \mu_k R_k^{\frac{2}{\theta}} = 2^{-k}\mu_0 R_0^{\frac{2}{\theta}}. \tag{38}$$

*Moreover, for $k > k^*$ there is a Borel set $\mathcal{C}_k \subset \Omega$ of probability at least $1 - (k - k^*)\bar{\delta}$ such that on $\mathcal{C}_k$, we have*

$$P(\widehat{\mathbf{w}}_k) - P(\widehat{\mathbf{w}}_{k^*}) \leq \mu_{k^*} R_{k^*}^{\frac{2}{\theta}}. \tag{39}$$

*Proof.* (Proof of Lemma 8) We prove (37) and (38) by induction. Note that (37) holds for $k = 1$. Assume it is true for some $k > 1$ on $\mathcal{A}_{k-1}$. According to the Proposition 1, there exists a Borel set $\mathcal{B}_k$ with $\Pr(\mathcal{B}_k) \geq 1 - \bar{\delta}$ such that

$$P(\widehat{\mathbf{w}}_k) - P_* \leq R_{k-1}G\left(\frac{1}{\sqrt{n_0+1}} + \frac{4\sqrt{2\log(2/\bar{\delta})}}{\sqrt{n_0+1}}\right) = R_{k-1}a(n_0, \bar{\delta})$$

$$= \frac{1}{2}\mu_k 2^{(1-\frac{2}{\theta})k}R_0^{\frac{2}{\theta}-1}R_{k-1} = \mu_k R_k^{\frac{2}{\theta}},$$

which is (38). By the inductive hypothesis, $\|\widehat{\mathbf{w}}_{k-1} - \mathbf{w}_{k-1}^*\|_2 \leq R_{k-1}$ on the set $\mathcal{A}_{k-1}$. Define $\mathcal{A}_k = \mathcal{A}_{k-1} \cap \mathcal{B}_k$. Note that

$$\Pr(\mathcal{A}_k) \geq \Pr(\mathcal{A}_{k-1}) + \Pr(\mathcal{B}_k) - 1 \geq 1 - k\bar{\delta},$$

and on $\mathcal{A}_k$, by the EBC and the definition of $k^*$, we have

$$\|\widehat{\mathbf{w}}_k - \widehat{\mathbf{w}}_k^*\|_2^{\frac{2}{\theta}} \leq \alpha^{\frac{1}{\theta}}(P(\widehat{\mathbf{w}}_k) - P_*) \leq \frac{P(\widehat{\mathbf{w}}_k) - P_*}{\mu_{k^*}} \leq \frac{\mu_k R_k^{\frac{2}{\theta}}}{\mu_{k^*}} \leq R_k^{\frac{2}{\theta}},$$

which is (37) for $k + 1$.

Now we prove (39). For $k > k^*$, by Lemma 7, there exists a Borel set $\mathcal{B}_k$ with $\Pr(\mathcal{B}_k) \geq 1 - \bar{\delta}$ such that

$$P(\widehat{\mathbf{w}}_k) - P(\widehat{\mathbf{w}}_{k-1}) \leq \frac{\gamma_k G^2}{2} + \frac{4GR_{k-1}\sqrt{2\log(2/\delta)}}{\sqrt{n_0 + 1}} \leq R_{k-1}a(n_0, \bar{\delta}) = 2^{k^*-k}R_{k^*-1}a(n_0, \bar{\delta})$$
$$= 2^{k^*-k}\mu_{k^*}R_{k^*}^{\frac{2}{\theta}} = \mu_k R_k^{\frac{2}{\theta}},$$

which implies that on $\mathcal{C}_k = \cap_{j=k^*+1}^{k}\mathcal{B}_j$, we have

$$P(\widehat{\mathbf{w}}_k) - P(\widehat{\mathbf{w}}_{k^*}) = \sum_{j=k^*+1}^{k}(P(\widehat{\mathbf{w}}_j) - P(\widehat{\mathbf{w}}_{j-1})) \leq \sum_{j=k^*+1}^{k}2^{k^*-j}\mu_{k^*}R_{k^*}^{\frac{2}{\theta}} \leq \mu_{k^*}R_{k^*}^{\frac{2}{\theta}}.$$

By union bound, we have $\Pr(\cap_{j=k^*+1}^{k}\mathcal{B}_j) \geq 1 - (k - k^*)\bar{\delta}$. Here completes the proof.

$\square$

Now we proceed the proof as follows. Note that $\mu_0 \leq \alpha^{-\frac{1}{\theta}} \leq \mu_m$. At the end of $k^*$-th stage, on the Borel set $\mathcal{A}_{k^*}$ of probability at least $1 - k^*\bar{\delta}$, we have

$$P(\widehat{\mathbf{w}}_{k^*}) - P_* \leq \mu_{k^*}R_{k^*}^{\frac{2}{\theta}}.$$

Then on the Borel set $\mathcal{D}_m = \mathcal{C}_m \cap \mathcal{A}_{k^*} = (\cap_{j=k^*+1}^{m}\mathcal{B}_j) \cap \mathcal{A}_{k^*}$ with $\Pr(\mathcal{D}_m) \geq 1 - m\bar{\delta}$, we have

$$P(\widehat{\mathbf{w}}_m) - P_* = P(\widehat{\mathbf{w}}_m) - P(\widehat{\mathbf{w}}_{k^*}) + (P(\widehat{\mathbf{w}}_{k^*}) - P_*) \leq 2\mu_{k^*}R_{k^*}^{\frac{2}{\theta}} \leq 4\left(\frac{\mu_{k^*}}{\alpha^{-\frac{1}{\theta}}}\right)^{\frac{1}{\theta}-1}\mu_{k^*}R_{k^*}^{\frac{2}{\theta}}$$

$$= 4\left(\frac{2^{(\frac{2}{\theta}-1)k^*}\mu_0}{\alpha^{-\frac{1}{\theta}}}\right)^{\frac{1}{\frac{2}{\theta}-1}}\mu_{k^*}R_{k^*}^{\frac{2}{\theta}} = 4(2^{k^*}\mu_{k^*}R_{k^*}^{\frac{2}{\theta}}\mu_0^{\frac{\theta}{2-\theta}}\alpha^{\frac{1}{2-\theta}})$$

$$= 4(\mu_0 R_0^{\frac{2}{\theta}}\mu_0^{\frac{\theta}{2-\theta}}\alpha^{\frac{1}{2-\theta}}) = 4[(2R_0^{1-\frac{2}{\theta}}a(n_0, \bar{\delta}))^{\frac{2}{2-\theta}}R_0^{\frac{2}{\theta}}\alpha^{\frac{1}{2-\theta}}] = 4(2\sqrt{\alpha} \cdot a(n_0, \bar{\delta}))^{\frac{2}{2-\theta}}$$

$$= (2^{2-\theta}2\sqrt{\alpha} \cdot a(n_0, \bar{\delta}))^{\frac{2}{2-\theta}}.$$

By the definition of $m$ and $\bar{\delta}$, and the fact that $m \leq \frac{1}{2}\log_2 n$, we have $m\bar{\delta} \leq \delta$. So $\Pr(\mathcal{D}_m) \geq 1 - \delta$.

**Case 2.** If $\alpha^{-\frac{1}{\theta}} < \mu_0$, then on $\mathcal{A}_1 = \mathcal{B}_1$,

$$P(\widehat{\mathbf{w}}_1) - P_* \leq R_0 \cdot a(n_0, \bar{\delta}) = \frac{R_0}{a(n_0, \bar{\delta})^{\frac{\theta}{2-\theta}}} \cdot a(n_0, \bar{\delta})^{\frac{2}{2-\theta}}$$

$$= \frac{2^{\frac{\theta}{2-\theta}}}{\mu_0^{\frac{\theta}{2-\theta}}}a(n_0, \bar{\delta})^{\frac{2}{2-\theta}} \leq 2^{\frac{\theta}{2-\theta}}\left(\sqrt{\alpha} \cdot a(n_0, \bar{\delta})\right)^{\frac{2}{2-\theta}}.$$

Hence on $\mathcal{A}_1 \cap \mathcal{C}_m$, by a similar argument as in case 1, we have

$$P(\widehat{\mathbf{w}}_m) - P_* = P(\widehat{\mathbf{w}}_m) - P(\widehat{\mathbf{w}}_1) + P(\widehat{\mathbf{w}}_1) - P_* \leq 2R_0 \cdot a(n_0, \bar{\delta}) \leq (2\sqrt{\alpha} \cdot a(n_0, \bar{\delta}))^{\frac{2}{2-\theta}},$$

where $\Pr(\mathcal{A}_1 \cap \mathcal{C}_m) \geq 1 - \delta$.

Combining the two cases, we have with probability at least $1 - \delta$,
$P(\widehat{\mathbf{w}}_m) - P_*$

$$\leq (8\sqrt{\alpha} \vee 2\sqrt{\alpha})^{\frac{2}{2-\theta}}\left(G\left(\frac{1}{\sqrt{n_0 + 1}} + \frac{4\sqrt{2\log(2/\bar{\delta})}}{\sqrt{n_0 + 1}}\right)\right)^{\frac{2}{2-\theta}}$$

$$\leq (64\alpha)^{\frac{1}{2-\theta}}\left(\frac{G\left(1 + 4\sqrt{2\log(\frac{\log_2 n}{\delta})}\right)}{\sqrt{\frac{n}{\frac{1}{2}\log_2 n}}}\right)^{\frac{2}{2-\theta}} = \left(\frac{128\alpha G^2 \log_2 n \left(1 + 4\sqrt{2\log(\frac{\log_2 n}{\delta})}\right)^2}{n}\right)^{\frac{1}{2-\theta}},$$

where the second inequality stems from the fact that $n_0 + 1 \geq \frac{n}{m} \geq \frac{n}{\frac{1}{2}\log_2 n}$.

$\square$

# H  Detailed Analysis of Examples Satisfying EBC

**Risk Minimization Problems over an $\ell_2$ ball.**

**Lemma 9.** *Consider the following problem*

$$\min_{\|\mathbf{w}\|_2 \leq B} P(\mathbf{w}) \triangleq \mathbb{E}_{\mathbf{z}}[f(\mathbf{w}, \mathbf{z})] \tag{40}$$

*If* $\min_{\mathbf{w} \in \mathbb{R}^d} P(\mathbf{w}) < \min_{\|\mathbf{w}\|_2 \leq B} P(\mathbf{w})$*, then the above problem satisfies* $EBC(\theta = 1, \alpha)$*.*

*Proof.* The proof is similar to that of Theorem 3.5 of [24]. Denote $\mathbf{w}_*$ by an optimal solution of Example 4. Let $\Omega = \{\mathbf{w} \in \mathbb{R}^d \mid \|\mathbf{w}\|_2 \leq B\}$, and $F(\mathbf{w}) = P(\mathbf{w}) + I_\Omega(\mathbf{w})$, where $I_\Omega(\mathbf{w}) = 0$ if $\mathbf{w} \in \Omega$, and otherwise $I_\Omega(\mathbf{w}) = +\infty$. Then we have $\arg\min_{\mathbf{w} \in \mathbb{R}^d} F(\mathbf{w}) = \arg\min_{\|\mathbf{w}\|_2 \leq B} P(\mathbf{w})$. Let $\mathbf{w}_* \in \arg\min_{\mathbf{w} \in \mathbb{R}^d} F(\mathbf{w})$ denote an optimal solution.

Since $B > 0$, so the optimization problem is strictly feasible, then by the Lagrangian theory, there exists some $\lambda \geq 0$, such that

$$F(\mathbf{w}_*) = \min_{\|\mathbf{w}\|_2 \leq B} P(\mathbf{w}) = \min_{\mathbf{w} \in \mathbb{R}^d} (P(\mathbf{w}) + \lambda(\|\mathbf{w}\|_2^2 - B^2))$$

$$= P(\mathbf{w}_*) + \lambda(\|\mathbf{w}_*\|_2^2 - B^2).$$

Note that $\min_{\mathbf{w} \in \mathbb{R}^d} P(\mathbf{w}) < \min_{\|\mathbf{w}\|_2 \leq B} P(\mathbf{w})$, as a result $\lambda > 0$. Then by complementary slackness, we know that $\|\mathbf{w}_*\|_2 = B$. Denote by $P_\lambda(\mathbf{w}) = P(\mathbf{w}) + \lambda(\|\mathbf{w}\|_2^2 - B^2)$. Then according to Theorem 28.1 [34], we have

$$\mathbf{w}_* \in \arg\min F = \{\mathbf{w} \mid \|\mathbf{w}\|_2 = B\} \cap \arg\min_{\mathbf{w} \in \mathbb{R}^d} P_\lambda(\mathbf{w}). \tag{41}$$

Since $P_\lambda(\mathbf{w})$ is strongly convex due to $\lambda > 0$, its optimal solution is unique. As a result,

$$\mathbf{w}_* = \arg\min F = \arg\min_{\mathbf{w} \in \mathbb{R}^d} P_\lambda(\mathbf{w}). \tag{42}$$

In addition, there exists $\mu > 0$ such that (due to the strong convexity of $P_\lambda(\mathbf{w})$),

$$\|\mathbf{w} - \arg\min P_\lambda(\mathbf{w})\|_2 \leq \mu(P_\lambda(\mathbf{w}) - \min_{\mathbf{w}} P_\lambda(\mathbf{w}))^{1/2}$$

$$= \mu(P(\mathbf{w}) + \lambda(\|\mathbf{w}\|_2^2 - B^2) - P(\mathbf{w}_*))^{1/2}$$

$$\leq \mu(P(\mathbf{w}) - P(\mathbf{w}_*))^{1/2}.$$

Then according to (42), we know that

$$\|\mathbf{w} - \mathbf{w}_*\|_2^2 \leq \mu^2(P(\mathbf{w}) - P(\mathbf{w}_*)),$$

which is $EBC(\theta = 1, \mu^2)$. $\qquad\square$

**Quadratic Problems.**

**Lemma 10.** *Consider the following problem*

$$\min_{\mathbf{w} \in \mathcal{W}} P(\mathbf{w}) \triangleq \mathbf{w}^\top \mathbb{E}_{\mathbf{z}}[A(\mathbf{z})]\mathbf{w} + \mathbf{w}^\top \mathbb{E}_{\mathbf{z}'}[\mathbf{b}(\mathbf{z}')] + c \tag{43}$$

*If* $\mathbb{E}_{\mathbf{z}}[A(\mathbf{z})]$ *is PSD and* $\mathcal{W}$ *is a bounded polyhedron, then the above problem satisfies* $EBC(\theta = 1, \alpha)$*.*

*Proof.* Let us consider $\mathbb{E}_{\mathbf{z}}[A(\mathbf{z})] \neq 0$; otherwise it reduces to PLP.

Note that $\mathbb{E}_{\mathbf{z}}[A(\mathbf{z})]$ is PSD, so there exists a nonzero matrix $A$ such that $\mathbb{E}_{\mathbf{z}}[A(\mathbf{z})] = A^\top A$. The original optimization problem is equivalent to

$$\min_{\mathbf{w} \in \mathcal{W}} g(A\mathbf{w}) + \mathbf{w}^\top \mathbb{E}_{\mathbf{z}'}[b(\mathbf{z}')] + c, \tag{44}$$

where $g(\mathbf{u}) = \mathbf{u}^\top \mathbf{u}$ is a strongly convex function of $\mathbf{u}$. Since the constraint is a polyhedral function of $\mathbf{w}$, according to the Lemma 12 of [52], we know that the optimization problem satisfies $EBC(\theta = 1, \alpha)$.

$\qquad\square$

**Piecewise Linear Problems (PLP)**

**Lemma 11.** *Consider the problem*

$$\min_{\mathbf{w} \in \mathcal{W}} P(\mathbf{w}) \triangleq \mathbb{E}[f(\mathbf{w}, \mathbf{z})] \tag{45}$$

*where $\mathbb{E}[f(\mathbf{w}, \mathbf{z})]$ is a piecewise linear function and $\mathcal{W}$ is a bounded polyhedron. Then the problem (45) satisfies $EBC(\theta = 1, \alpha)$.*

*Proof.* According to weak sharp minima condition [5] (e.g., Lemma 8 in [52]), we have
$$\|\mathbf{w} - \mathbf{w}^*\|_2^2 \le c(P(\mathbf{w}) - P(\mathbf{w}^*))^2,$$
Since $P(\mathbf{w})$ is piecewise linear, then $P(\mathbf{w}) - P(\mathbf{w}_*)$ is bounded on a bounded set. Then there exists $\alpha > 0$ such that
$$\|\mathbf{w} - \mathbf{w}^*\|_2^2 \le \alpha(P(\mathbf{w}) - P(\mathbf{w}^*)),$$

$\square$

### $\ell_1$ regularized problems

**Lemma 12.** *Consider the problem: for $\ell_1$ regularized risk minimization:*
$$\min_{\|\mathbf{w}\|_1 \le B} F(\mathbf{w}) \triangleq P(\mathbf{w}) + \lambda\|\mathbf{w}\|_1, \tag{46}$$
*If $P(\mathbf{w})$ is convex quadratic or piecewise linear, then the problem (46) satisfies $EBC(\theta = 1, \alpha)$.*

*Proof.* It is easy to see that $P(\mathbf{w})$ is either piecewise linear or piecewise convex quadratic. According to Lemma 3.3 of [23], we have

- When $P(\mathbf{w})$ is piecewise linear, there exists $\alpha_1, \alpha > 0$, such that
$$\|\mathbf{w} - \mathbf{w}^*\|_2^2 \le \alpha_1(P(\mathbf{w}) - P(\mathbf{w}^*))^2 \le \alpha(P(\mathbf{w}) - P(\mathbf{w}^*)),$$
  where we use the fact $P(\mathbf{w}) - P(\mathbf{w}_*)$ is bounded over a bounded domain due to its Lipschitz continuity.

- When $P(\mathbf{w})$ is piecewise convex quadratic, there exists $\alpha_2 > 0$, such that
$$\|\mathbf{w} - \mathbf{w}^*\|_2^2 \le \alpha_2(P(\mathbf{w}) - P(\mathbf{w}^*)).$$

$\square$

**Lemma 13.** *Consider the problem:*
$$\min_{\mathbf{w} \in \mathcal{W}} F(\mathbf{w}) \triangleq P(\mathbf{w}) + \lambda\|\mathbf{w}\|_p^p \tag{47}$$
*If $P(\mathbf{w})$ is convex quadratic, and $\mathcal{W}$ is a bounded polyheron, then the above problem satisfies $EBC(\theta = 1/p, \alpha)$.*

*Proof.* According to Theorem 5.2 [53], the objective function is $p$-th order convex polynomial function and $\forall \mathbf{w} \in \mathcal{W}$ there exists $\tau > 0$ such that
$$\|\mathbf{w} - \mathbf{w}^*\|_2 \le \tau(P(\mathbf{w}) - P(\mathbf{w}^*) + (P(\mathbf{w}) - P(\mathbf{w}^*))^{1/p}).$$
There exists $c > 0$ such that $P(\mathbf{w}) - P(\mathbf{w}^*) \le c$ for any $\mathbf{w} \in \mathcal{W}$. Then
$$\|\mathbf{w} - \mathbf{w}^*\|_2 \le \tau(c^{1-1/p} + 1)(P(\mathbf{w}) - P(\mathbf{w}^*))^{1/p},$$
i.e.,
$$\|\mathbf{w} - \mathbf{w}^*\|_2^2 \le \tau^2(c^{1-1/p} + 1)^2(P(\mathbf{w}) - P(\mathbf{w}^*))^{2/p}.$$

$\square$

## I  A Variant of ASA without projection into intersection

Now we provide a different variant of ASA, which utilizes SSGS (Algorithm 2 in [49]) as a subroutine to avoid the projection onto the intersection of $\mathcal{W}$ and a bounded ball in the vanilla ASA. SSGS is an algorithm which adds a strongly convex regualarizer to the original loss function, i.e.,
$$\min_{\mathbf{w} \in \mathcal{W}} P(\mathbf{w}) + \frac{1}{2\beta}\|\mathbf{w} - \mathbf{w}_1\|_2^2,$$
where $\mathbf{w}_1 \in \mathcal{W}$ is called reference point. For completeness, we describe the SSGS and the corresponding ASA2 algorithms in Algorithm 3 and Algorithm 4 respectively.

We first present a result for analyzing SSGS, which is the Corollary 5 in [49].

| **Algorithm 3** SSGS($\mathbf{w}_1, \beta, T$) | **Algorithm 4** ASA2($\mathbf{w}_1, n, R_0$) |
|---|---|
| **Input:** $\mathbf{w}_1 \in \mathcal{W}$, $\beta > 0$ and $T$ | **Input:** $\mathbf{w}_1 \in \mathcal{W}$, $n$ and $R_0 = 2R$ |
| **Output:** $\widehat{\mathbf{w}}_T$ | **Output:** $\widehat{\mathbf{w}}_m$ |
| 1: **for** $t = 1, \ldots, T$ **do** | 1: Set $\widehat{\mathbf{w}}_0 = \mathbf{w}_1$, $m = \lfloor \frac{1}{2} \log_2 \frac{2n}{\log_2 n} \rfloor - 1$, $n_0 =$ |
| 2: $\quad \mathbf{w}'_{t+1} = (1 - \frac{2}{t})\mathbf{w}_t + \frac{2}{t}\mathbf{w}_1 - \frac{2\beta}{t} g_t$ | $\quad \lfloor n/m \rfloor$ |
| 3: $\quad \mathbf{w}_{t+1} = \Pi_{\mathcal{W}}(\mathbf{w}'_{t+1})$ | 2: **for** $k = 1, \ldots, m$ **do** |
| 4: **end for** | 3: $\quad$ Set $\beta_k = \frac{R_{k-1}\sqrt{n_0}}{2G}$ and $R_k = R_{k-1}/2$ |
| 5: $\widehat{\mathbf{w}}_T = \frac{1}{T+1} \sum_{t=1}^{T+1} \mathbf{w}_t$ | 4: $\quad \widehat{\mathbf{w}}_k = \text{SSGS}(\widehat{\mathbf{w}}_{k-1}, \beta_k, n_0)$ |
| 6: return $\widehat{\mathbf{w}}_T$ | 5: **end for** |

**Proposition 2.** *Suppose* **Assumptions 1 and 2** *hold. Let $0 < \delta < 1/e$, $T \geq 3$, $\mathbf{w}^* \in \mathcal{W}_*$ be the closest optimal solution to $\mathbf{w}_1$, and $R_0$ be an upper bound on $\|\mathbf{w}_1 - \mathbf{w}^*\|_2$. Apply $T$ iterations of the SSGS (Algorithm 3) and return the average solution, where $g_t$ is a stochastic subgradient of $P(\mathbf{w})$ at $\mathbf{w}_t$. With probability at least $1 - \delta$, we have*

$$P(\widehat{\mathbf{w}}_T) - P_* \leq \frac{1}{2\beta}\|\mathbf{w}_1 - \mathbf{w}^*\|_2^2 + \frac{34\beta G^2(1 + \log T + \log(4\log T/\delta))}{T}.$$

*where $\widehat{\mathbf{w}}_T = \frac{1}{T+1}\sum_{t=1}^{T+1}\mathbf{w}_t$. Moreover, choose $\beta = \frac{R_0\sqrt{T}}{2G}$, and then with probability at least $1 - \delta$,*

$$P(\widehat{\mathbf{w}}_T) - P_* \leq R_0 G\left(\frac{1}{\sqrt{T}} + \frac{17(1 + \log T + \log(4\log T/\delta))}{\sqrt{T}}\right).$$

*Similarly, for any nonnegative $R_0$, by choosing $\beta = \frac{R_0\sqrt{T}}{2G}$, and then with probability at least $1 - \delta$,*

$$P(\widehat{\mathbf{w}}_T) - P(\mathbf{w}_1) \leq R_0 G\left(\frac{1}{\sqrt{T}} + \frac{17(1 + \log T + \log(4\log T/\delta))}{\sqrt{T}}\right).$$

Then we provide the high probability analysis of ASA2, which is Theorem 5.

**Theorem 5.** *Suppose Assumptions 1, and 2 hold. Let $\widehat{\mathbf{w}}_m$ be the returned solution of the Algorithm 4. For $n \geq 100$ and any $\delta \in (0,1)$, with probability at least $1 - \delta$, we have*

$$P(\widehat{\mathbf{w}}_m) - P_* \leq O\left(\frac{\alpha G^2 \log(n)(\log n + \log(\frac{\log n}{\sqrt{\delta}}))^2}{n}\right)^{\frac{1}{2-\theta}}.$$

*Proof.* We use the same notation as that in the proof of Theorem 3 unless specified. Define

$$a(n, \bar{\delta}) = G\left(\frac{1}{\sqrt{n}} + \frac{17(1 + \log n + \log(4\log n/\bar{\delta}))}{\sqrt{n}}\right). \tag{48}$$

First we show that when $n \geq 100$, we have

$$\frac{1}{2}\sqrt{\frac{2n}{\log_2 n}}\left(\frac{1}{\sqrt{n_0}} + \frac{17(1 + \log n_0 + \log(4\log n_0/\bar{\delta}))}{\sqrt{n_0}}\right) \geq 1.$$

Note that

$$\text{LHS} \geq \sqrt{\frac{2n}{\log_2 n}}\left(\frac{\sqrt{17(1 + \log n_0 + \log(4\log n_0/\bar{\delta}))}}{\sqrt{n_0}}\right)$$

$$\geq \sqrt{\frac{34m(1 + \log(\frac{n}{m} - 1) + \log(4\log(\frac{n}{m} - 1)/\bar{\delta}))}{\log_2 n}}$$

$$\geq \sqrt{\frac{17(\log_2 n - \log_2 \log_2 n - 3) \cdot \mathcal{F}_1}{\log_2 n}}$$

$$\geq \sqrt{17(1 - \frac{\log_2 \log_2 n + 3}{\log_2 n})} \geq 1 = \text{RHS},$$

where $\mathcal{F}_1 = (1 + \log(\frac{n}{m} - 1) + \log(2\log(\frac{n}{m} - 1)\log_2 n/\delta))$. The first inequality holds by utilizing the fact that $a + b \geq 2\sqrt{ab}$, the second inequality holds since $n \geq 100$, and then $3 \leq \frac{n}{m} - 1 \leq n_0 = \lfloor \frac{n}{m} \rfloor \leq \frac{n}{m}$, the third inequality holds because of $m \geq \frac{1}{2}\log_2 \frac{2n}{\log_2 n} - 2 > 0$ and definition of $\bar{\delta}$, the fourth and fifth inequalities hold since $n \geq 100$ and $m \leq \frac{1}{2}\log_2 n$.

---

**Algorithm 5** PSG($\mathbf{w}_1, \gamma, T, \mathcal{W}$)

---

**Input:** $\mathbf{w}_1 \in \mathcal{W}$, $\gamma > 0$ and $T$
**Output:** $\widehat{\mathbf{w}}_T$
1: **for** $t = 1, \ldots, T$ **do**
2:     Compute

$$\mathbf{w}_{t+1} = \arg\min_{\mathbf{w} \in \mathcal{W}} \frac{1}{2}\|\mathbf{w} - \mathbf{w}_t\|_2^2 + \eta g_t^\top \mathbf{w} + \eta r(\mathbf{w}),$$

    where $g_t$ is the stochastic subgradient of $\mathbb{E}_{\mathbf{z} \sim \mathbb{P}}[f(\mathbf{w}, \mathbf{z})]$ evaluated at $\mathbf{w}_t$
3: **end for**
4: $\widehat{\mathbf{w}}_T = \frac{1}{T}\sum_{t=1}^T \mathbf{w}_t$
5: return $\widehat{\mathbf{w}}_T$

---

**Algorithm 6** ASA3($\mathbf{w}_1, n, R_0$)

---

**Input:** $\mathbf{w}_1 \in \mathcal{W}$, $n$ and $R_0 = 2R$
**Output:** $\widehat{\mathbf{w}}_m$
1: Set $\widehat{\mathbf{w}}_0 = \mathbf{w}_1$, $m = \lfloor \frac{1}{2}\log_2 \frac{2n}{\log_2 n}\rfloor - 1$, $n_0 = \lfloor n/m \rfloor$
2: **for** $k = 1, \ldots, m$ **do**
3:     Set $\gamma_k = \frac{R_{k-1}}{G\sqrt{n_0}}$ and $R_k = R_{k-1}/2$
4:

$$\widehat{\mathbf{w}}_k = \text{PSG}(\widehat{\mathbf{w}}_{k-1}, \gamma_k, n_0, \mathcal{W} \cap \mathcal{B}(\widehat{\mathbf{w}}_{k-1}, R_{k-1}))$$

5: **end for**
6: return $\widehat{\mathbf{w}}_m$

---

We can duplicate the rest of the proof of Theorem 3 other than using the definition of $a(n_0, \bar{\delta})$ according to (48). Finally, we have with probablity at least $1 - \delta$,

$$P(\widehat{\mathbf{w}}_m) - P_* \leq (64\alpha)^{\frac{1}{2-\theta}} a(n_0, \bar{\delta})^{\frac{2}{2-\theta}} \leq \left( \frac{64\alpha G^2(1 + 17\mathcal{F}_2)^2}{\frac{2n}{\log_2 n} - 1} \right)^{\frac{1}{2-\theta}},$$

where

$$\mathcal{F}_2 = 1 + \log\left( \frac{n}{\frac{1}{2}\log_2 \frac{2n}{\log_2 n} - 2} \right) + \log\left( 2\log\left( \frac{n}{\frac{1}{2}\log_2 \frac{2n}{\log_2 n} - 2} \right) \log_2 n/\delta \right).$$

The second inequality holds since $n_0 = \lfloor \frac{n}{m} \rfloor \geq \frac{n}{m} - 1$, $\frac{1}{2}\log_2 \frac{2n}{\log_2 n} - 2 \leq m \leq \frac{1}{2}\log_2 n$. $\qquad\square$

## J   A variant of ASA with a subroutine using proximal mapping

In this section, we consider the nonsmooth composite optimization problem (2), which is

$$\min_{\mathbf{w} \in \mathcal{W}} P(\mathbf{w}) \triangleq \mathbb{E}_{\mathbf{z} \sim \mathbb{P}}[f(\mathbf{w}, \mathbf{z})] + r(\mathbf{w}).$$

We introduce a variant of ASA, i.e., ASA3 (Algorithm 6), with a theoretical guarantee. ASA3 is a multistage scheme of proximal SGD (Algorithm 5).

Before analysis, we first present a standard result of proximal SGD, which is the Lemma 5 of [49].

**Proposition 3.** *Suppose* **Assumptions 1 and 2** *hold. In addition, we assume the proximal mapping in terms of $r(\mathbf{w})$ has a closed form, and $r(\mathbf{w})$ is $\rho$-Lipschitz continuous for any $\mathbf{w} \in \mathcal{W}$. Let $\epsilon \geq 0$ and $D$ be the upper bound of $\|\mathbf{w}_1 - \mathbf{w}_{1,\epsilon}^\dagger\|_2$, where $\mathbf{w}_{1,\epsilon}^\dagger$ is the point closed to $\epsilon$-sublevel set of $P(\mathbf{w})$. Denote $g_t$ by the stochastic subgradient of $\mathbb{E}_{\mathbf{z} \sim \mathbb{P}}[f(\mathbf{w}, \mathbf{z})]$ at $\mathbf{w}_t$. Apply $T$-iterations of the following steps:*

$$\mathbf{w}_{t+1} = \arg\min_{\mathbf{w} \in \mathcal{W} \cap \mathcal{B}(\mathbf{w}_1, D)} \frac{1}{2}\|\mathbf{w} - \mathbf{w}_t\|_2^2 + \eta g_t^\top \mathbf{w} + \eta r(\mathbf{w}).$$

*Given $\mathbf{w}_1$, for any $\delta \in (0, 1)$, we have with probability at least $1 - \delta$,*

$$P(\widehat{\mathbf{w}}_T) - P(\mathbf{w}_{1,\epsilon}^\dagger) \leq \frac{\eta G^2}{2} + \frac{\|\mathbf{w}_1 - \mathbf{w}_{1,\epsilon}^\dagger\|_2^2}{2\eta T} + \frac{4GD\sqrt{3\log(1/\delta)}}{\sqrt{T}} + \frac{\rho D}{T},$$

*where $\widehat{\mathbf{w}}_T = \frac{1}{T}\sum_{t=1}^T \mathbf{w}_t$.*

**Theorem 6.** *Suppose* **Assumptions 1 and 2** *hold. In addition, we assume the proximal mapping in terms of $r(\mathbf{w})$ has a closed form, and $r(\mathbf{w})$ is $\rho$-Lipschitz continuous for any $\mathbf{w} \in \mathcal{W}$. $\|\mathbf{w}_1 - \mathbf{w}^*\|_2 \le R_0$, where $\mathbf{w}^*$ is the closest optimal solution to $\mathbf{w}_1$. For $n \ge 100$, $n_0 \ge \frac{\rho^2}{G^2}$ and any $\delta \in (0, 1)$, with probability at least $1 - \delta$, the Algorithm ASA3 guarantees that*

$$P(\widehat{\mathbf{w}}_m) - P_* \le O\left( \frac{\bar{\alpha}(\log(n)\log(\log(n)/\delta))}{n} \right)^{\frac{1}{2-\theta}}.$$

*where $\bar{\alpha} = \max(\alpha G^2, (R_0 G)^{2-\theta})$.*

*Proof.* At first we derive the parallel version of the Proposition 1 and Lemma 7 in the case of solving problem (2), which is not difficult by utilizing the Proposition 3.

- We first prove the parallel version of the Proposition 1. By taking $\epsilon = 0$, then $\mathbf{w}_{1,\epsilon}^{\dagger}$ is the projection of $\mathbf{w}_1$ onto the optimal set $\mathcal{W}_*$, and we define it to be $\mathbf{w}^*$. If $R_0$ is a upper bound of $\|\mathbf{w}_1 - \mathbf{w}^*\|_2$, by taking $\eta = \frac{R_0}{G\sqrt{T}}$, then applying $T$ iterations of

$$\mathbf{w}_{t+1} = \underset{\mathbf{w} \in \mathcal{W} \cap \mathcal{B}(\mathbf{w}_1, R_0)}{\arg\min} \frac{1}{2}\|\mathbf{w} - \mathbf{w}_t\|_2^2 + \eta g_t^\top \mathbf{w} + \eta r(\mathbf{w})$$

has the guarantee that with probability at least $1 - \delta$,

$$P(\widehat{\mathbf{w}}_T) - P_* \le R_0 G \left( \frac{1}{\sqrt{T}} + \frac{4\sqrt{3\log(1/\delta)}}{\sqrt{T}} \right) + \frac{\rho R_0}{T}.$$

By choosing $T \ge \frac{\rho^2}{G^2}$, i.e., $\frac{\rho R_0}{T} \le \frac{R_0 G}{\sqrt{T}}$, and we have

$$P(\widehat{\mathbf{w}}_T) - P_* \le R_0 G \left( \frac{2}{\sqrt{T}} + \frac{4\sqrt{3\log(1/\delta)}}{\sqrt{T}} \right).$$

- We then prove the parallel version of the Lemma 7. We make choose $\epsilon$ large enough such that $\mathbf{w}_{1,\epsilon}^{\dagger} = \mathbf{w}_1$. By utilizing the Proposition 3, we know that for any nonnegative $R_0$, taking $\eta = \frac{R_0}{G\sqrt{T}}$ and applying $T$ iterations of

$$\mathbf{w}_{t+1} = \underset{\mathbf{w} \in \mathcal{W} \cap \mathcal{B}(\mathbf{w}_1, R_0)}{\arg\min} \frac{1}{2}\|\mathbf{w} - \mathbf{w}_t\|_2^2 + \eta g_t^\top \mathbf{w} + \eta r(\mathbf{w})$$

have the guarantee that with probability at least $1 - \delta$,

$$P(\widehat{\mathbf{w}}_T) - P(\mathbf{w}_1) \le R_0 G \left( \frac{1}{\sqrt{T}} + \frac{4\sqrt{3\log(1/\delta)}}{\sqrt{T}} \right) + \frac{\rho R_0}{T}.$$

By choosing $T \ge \frac{\rho^2}{G^2}$, i.e., $\frac{\rho R_0}{T} \le \frac{R_0 G}{\sqrt{T}}$, and we have

$$P(\widehat{\mathbf{w}}_T) - P_* \le R_0 G \left( \frac{2}{\sqrt{T}} + \frac{4\sqrt{3\log(1/\delta)}}{\sqrt{T}} \right).$$

The rest of the proof is similar to the proof of Theorem 3. $\square$

Finally, we mention that a stochastic mirror descent algorithm with a non-Euclidean norm prox-function can be used, e.g., the Composite Objective Mirror Descent algorithm with $p$-norm divergence in [8], Similar analysis based on Theorem 8 in [8] can be derived. When leveraging the error bound, we can use a $p$-norm version (i.e., changing the Euclidean norm to the $p$-norm and the corresponding parameter $\alpha$).