[Reviews · NeurIPS 2018]

Reviewer 1



I believe that the results are interesting given the relatively more general assumptions. I also liked the examples of Section 5 which more clearly show fast rates for some problems. I suggest the authors to include some ideas for future work in the current manuscript. === AFTER REBUTTAL === The authors only addressed Reviewer 4.

Reviewer 2



Summary: This paper developed faster rates of Empirical Risk Minimization (ERM) under Error Bound Conditions (EBC), and an efficient stochastic approximation (SA) algorithm that provides faster convergence compared to previous studies. Authors shows in-depth theoretical analysis on ERM under the condition and the SA algorithm and did an empirical experiments on various dataset to support the convergence rate of the SA algorithm. Quality: The technical content of the paper appears to be solid with various theoretical analyses and with experiments. I did enjoy the way it is organized from general concepts for ERM in Section 3, followed by SA algorithm in Section 4, and its application in Section 5. The logical flow made the paper very solid to follow and I feel this is really a solid piece of work besides of good contents in the paper. I also enjoyed reading Section 5 answering this essential question ‘why this research matters’. It applies the paper’s contributions in a various optimization problems and also shows empirical results of the suggested SA algorithm, which shows a solid technical achievements. Clarity: This paper is really well structured. I particularly liked the a very clear summary of the main contribution of the paper in the introduction section and its related work section. It did provide a great overview of the paper that the reader can focus on the core concepts. Originality: As pointed out in the related work section of this paper, learning under EBC was not explored intensively yet; thus making this paper novel. The suggested algorithm (SA) is also new to field with various strengths against other baseline methods. Significance: The paper showed solid contributions on this field as a new pioneer of ERM under EBC. Conclusion: I believe this is a solid paper, probably not the best paper though. -------------------------------------------------------------------------------------------- Post rebuttal: Thanks authors for clarifying concerns.

Reviewer 3



Overview: This paper develops fast excess risk bounds for ERM and develops stochastic optimization algorithms for losses that (at the population level) satisfy the so-called "error bound condition". The error bound condition generalizes a particular inequality implied by strong convexity to 1) non-convex but still curved losses 2) convex losses that are not strongly convex but enjoy similar (possibly slower) growth around the optimum. The authors give 1) An ERM guarantee that interpolates between \sqrt{d/n} and d/n as a function of the exponent for which the EBC holds, and does not assume convexity and 2) A faster/optimistic rate that depends on the loss of the benchmark, but requires convexity in addition to EBC 3) A stochastic optimization routine matching the (not optimistic) guarantee 1). Lastly, they instantiate their guarantees for a few simple families of losses. Originality and Significance: My takeaway is that the results in this paper do not require huge changes in proof technique compared to previous results that assume strong convexity/convexity (eg van Erven et al. "Fast rates in statistical and online learning" for Theorem 1 and Zhang et al., "Empirical risk minimization for stochastic convex optimization: O(1/n)- and o(1/n^2)-type of risk bounds" for Theorem 2), but I do like that the paper is fairly comprehensive and I think it will probably serve as a useful reference point for future research. Some further coments: * Theorem 2 is fairly restrictive since it has to assume convexity, but I see from the examples that there are indeed losses that satisfy EBC for theta > 0 yet are not strongly convex. * Theorem 3: To make the paper more comprehensive it would be better if this result could be extended to L*-dependent regime studied in Theorem 2. Also, please add more discussion as to why adapting to the parameter theta is a major technical challenge, Quality and Clarity: Overall I found they paper to be reasonably well-written and fairly easy to follow, though the intro has quite a few typos so please give it another pass. I do have some minor comments: * Line 42: Please be more precise on your assumptions on data and loss for the rates you give for ERM and SA. ERM can also get \sqrt{1/n} also depending on the assumptions on the data norm. * The PLP setup / (12) should apply to piecewise linear *convex* functions, correct? * Ass 2: Specify the norm wrt Lipschitzness is defined. * Line 107: The result of Srebro you mention can easily be adapted to the unknown L* case using the doubling trick, no? ------------------ After rebuttal: I am keeping my score, as the rebuttal only addresses minor technical comments and not the overall issue of novelty in analysis and proof techniques.